# Potential Transformative Changes in Water Provision Systems: Impact of Decentralised Water Systems on Centralised Water Supply Regime

**Diederik van Duuren** [1,2,*], **Henk-Jan van Alphen** [3], **Steven H. A. Koop** [3,4] **and Erwin de Bruin** [1]

1   Waterleiding Maatschappij Limburg (WML), Limburglaan 25, 6229 GA Maastricht, The Netherlands
2   International Centre for Integrated Assessment and Sustainable Development (ICIS), Maastricht University, Kapoenstraat 2, 6211 KR Maastricht, The Netherlands
3   KWR Watercycle Research Institute, Groningenhaven 7, 3433 PE Nieuwegein, The Netherlands
4   Coopernicus Institute of Sustainable Development Utrecht University, Heidelberglaan 2, 3584 CS Utrecht, The Netherlands
*   Correspondence: D.vanDuuren@wml.nl; Tel.: +31-6-1519-1070

**Abstract:** Sustainable Urban Water Management (SUWM) is a paradigm in which decentralisation is key. There has been little work directed towards the large-scale possibilities of decentralised water systems and their implications on the functioning of the centralised (potable) water system. This study includes both a historical and future (scenario) analysis of decentralised developments. Integrated morphological socio-technical scenarios are combined with quantitative water flows for a case study (the Province of Limburg, the Netherlands) and examined by a transdisciplinary group of experts. The study shows how SUWM measures which focus on climate adaptation and circularity can have a significant impact on existing centralised potable water systems. In turn, influencing the total water and peak demands and thus resulting in different utilisation rates. This can result in more system failures (e.g., longer residence time, bacterial growth, reduced self-cleaning capacity), significant changes in the centralised infrastructure (e.g., more wells), increasing water bills (e.g., inequalities), and the preservation of aquifers for future generation. Different scenarios either have regime-reproducing or regime-diversifying impacts. SUWM measures are studied in isolation and thus externalities are not fully considered. Therefore, when planning for decentralised SUWM solutions, a systems thinking approach is recommended, which takes into account externalities.

**Keywords:** decentralised water systems; sustainable urban water management; rainwater harvesting; centralised water systems; hybrid water systems; scenarios; foresight; transformative processes

## 1. Introduction

Since the 19th Century, centralised water and sewer infrastructures have been built to address and solve issues related to hygiene and have, therefore, resulted in a significant reduction of diseases [1]. Centralised water systems are characterised by large treatment facilities, a distribution network which connects distant water sources and households and a top-down governance model [2]. This system has been optimised by a myriad of incremental changes over the past decades. Most countries spend between 1% to 6% of their annual GDP on centralised water infrastructure [3], resulting in substantial sunk expenditures, full dependency on these water services and, consequently, a lock-in situation [4] in which transformative water management alternatives are impeded. However, the rigid system is currently operating within a fast paced and ever-changing environment. Which includes, climate-change-induced challenges such as increased rainfall abnormalities and heat waves [5]. At the

same time, this system must maintain its service delivery with increasing consumer standards at an affordable cost [6]. In turn, many studies have concluded that such a centralised water system does not possess the capacity to deal with the vast and diverse challenges that we as a society face, and seek for a new water management paradigm [7–13]. The traditional, predict-and-control water management paradigm is based on centralised and fragmented organisation of drinking water, stormwater, wastewater collection and treatment [7,8,10]. There is increasing support for a more integrated and adaptive management paradigm which emphasises decentralised system configurations (both technological- and nature-based) [7,8,10,11]. Such configurations are characterised by the inclusion of small-scale systems, based on local sources, and a multi-level governance model [2,7,10,14]. In this paper, we use the terminology 'Sustainable Urban Water Management' (SUWM), as stated by Marlow et al. [12] to describe this new water management paradigm.

SUWM measures are considered necessary to maintain and improve water service delivery for now and in the future. Examples of well-known decentralised SUWM measures include rainwater harvesting [15], water reclamation [16], grey water recycling [16], source separation [17] referred to as "new sanitation" [18], green and blue infrastructure [19], and a diverse range of household water saving technologies [20]. Additionally, pro-environmental behaviour campaigns are diversifying and on the increase [21]. The reasons for these developments are as diverse as the possibilities themselves. Ranging from resource and nutrient recovery (in wastewater) [18], adapting to extreme weather conditions (dry spells and extreme rain events) [15], political stability [16], reducing environmental impact [11], energy reduction and recovery [17], increasing the adaptive capacity of the aged centralised infrastructure [11] and more.

Most of the literature highlights the benefits and potential positive impacts of decentralised SUWM solutions in cities [11]. However, despite the benefits claimed by proponents, the large-scale adoption of decentralised systems has failed to go beyond the demonstration phase in most areas of the world [14]. Several studies identify a wide range of social-technical impediments which explain the slow adoption of these solutions [9,11,12,14,22,23], and describe how these can be overcome in multi-level governance systems [24] and develop tools to support decision-making [25]. Others have developed methods to accelerate the processes of replication, transfer and uptake for decentralised SUWM based on transition study features [26,27].

Therefore, it can be observed that a vast number of publications focus on 'stimulating' the adoption of a new water management paradigm. However, what is missing is a critical reflection of the potential impact—both positive and negative—of the large-scale adoption of SUWM solutions on existing centralised infrastructure [28–31].

Moreover, there is heavy emphasis placed on narrow and specific research related to fragmented parts of water systems and little research that takes a holistic systems approach [30]. Modifications in any physical, operational, and institutional part of the system impacts the performance of other parts and the entire systems performance [30]. This relates to the complex nature of water infrastructure. Agudelo-Vera et al. [6] consider this infrastructure as being an inherently socio–technical system. On the one hand, water infrastructure comprises of physical and technological components, such as distribution pipelines and treatment facilities, whilst on the other hand, it is shaped by and itself shapes social and organisational processes, including actors such as consumers, operators and managers. These different components are in continuous interaction and subject to external and internal pressures leading to small changes that could result in structural changes or transitions of the way a socio–technical system operates [6].

We argue that SUWM solutions have the potential to place both internal and external pressures on the current centralised system through a plethora of ways. For example, SUWM measures change water demand patterns and the utilisation rate of centralised infrastructure. Furthermore, there is an increased risk of contamination within a centralised system when decentralised systems that include other types of water quality are connected. The aforementioned changes impact the functioning of a centralised system. Additionally, knowledge about feedback loops, and unforeseen and unwanted

effects (externalities), on the centralised infrastructure of a large-scale introduction of decentralised water systems is limited. Therefore, in reference to the argument provided by Leigh and Lee [11] 'connectivity between different water sources, treatment facilities and distribution networks is needed to overcome the lack of flexibility and adaptability in conventional water systems', may also have counter effects.

Changing climatic and socio-demographic circumstances require a long-term assessment scope, with the lifespan of centralised systems often being 100 years or more. In turn, the planning horizon for such systems is long and complex, and requires large capital investments and related risks [23]. The increasing number of decentralised pilots provide a window of opportunity to integrate decentralised solutions which enhance adequate delivery of water services in long-term water infrastructure planning. However, we currently lack a sufficient number of case studies to understand the complex interaction of hybrid urban water systems [31]. In many predictions and foresight studies on water demand, SUWM measures are not taken into account [32–34]. It is, therefore, essential to address the following research question: *"What is the future potential for decentralised socio-technical water systems and how will this affect the existing centralised system?"*

In order to address this question, a local water provision area—the province of Limburg, the Netherlands—has been selected as a case study. Here, a brief historical transition analysis and an elaborate foresight study will be applied. In this elaborate foresight study, different socio-technical scenarios that explicitly include SUWM measures (e.g., rainwater systems, grey water systems, green gardens, and several water saving devices) have been developed and analysed by a transdisciplinary group of experts. With this research set-up, we intend to achieve a better understanding of transformative change in water systems and thus, contribute to the scientific literature, helping policy makers, asset owners and asset operators.

## 2. Methods

In order to answer the research question, there are two lines of research both containing their own theory, methodology and findings. Firstly, a historical transition analysis of the water system. Which includes upcoming niches, internal barriers and drivers, and external pressures (landscape) [35] with a focus on the Netherlands (Figure 1). The second line of research extends into the future, whereby a foresight study was developed and applied to the case study area. The conceptual framework is presented in Figure 2. The 11 steps which were taken are discussed in more detail in the remainder of this section. Only a summary of the first part of this research will be presented, whilst the main focus of this article is presented in the second part which focuses on future scenarios.

Although quantitative data is used, the argumentation line of this study is that of a qualitative nature. Bryman [36] states that qualitative empirical research tries to find evidence for argument generalisation rather than statistical proof. However, the potential for generalisation from a case study remains limited [37]. Nevertheless, this type of research results in more in-depth understanding and nuanced findings [36], which aligns with the aim of understanding the complexity of the interactions between decentralised and centralised water systems.

### 2.1. Historical Transition Analysis

The historical analysis of the water provision system in the Netherlands extends over a period of about 30 years (1990–2017). In this period, the number of water utilities declined from 52 in 1990 to only 10 by the year 2000 due to the merging of local (often municipal) utilities into regional utilities.

The underlying methodology used to analyse transformative processes is the Multi-Level Perspective (MLP), developed by Geels [38]. Transformative processes (or so-called transitions) exist within a series of changes, which reinforce each other, and result in a changing societal system [39]. To define systems from this social perspective, Geels [1] points out that: *'artefacts by themselves have no power, . . . only in association with human agency and social structures and organisations do artefacts fulfil functions'*. A combination of 'the social' and 'the technical' is needed to analyse functional

artefacts, such as the water provisioning system. Hence, socio-technical transitions are co-evolutionary changing processes over several domains and scales [40] ranging from Social/cultural, Economic, Political/institutional, Technological, Ecological, [41] and Demographic [6] processes and developments (abbreviated as SEPTED developments) on various levels of scale.

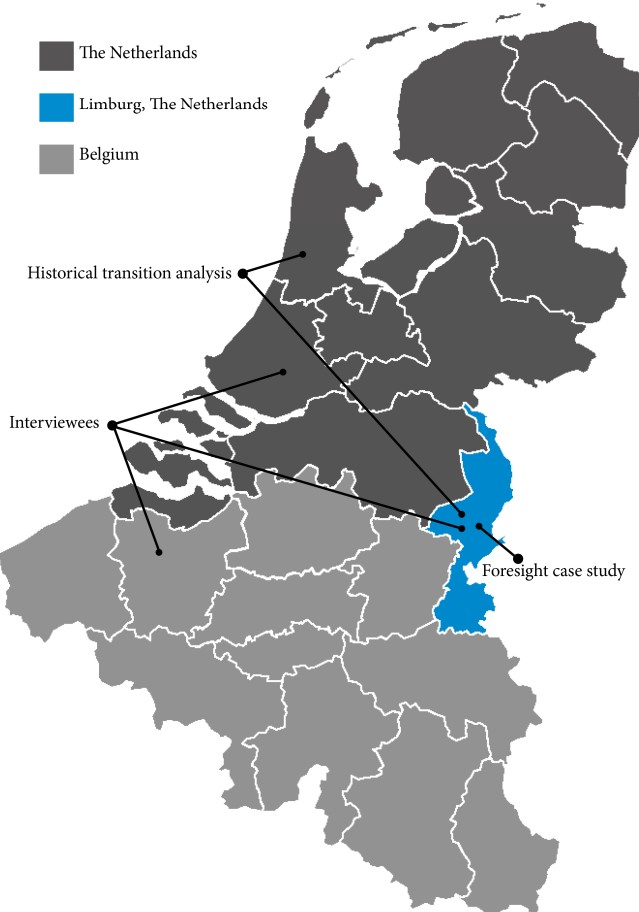

**Figure 1.** Geographical context of the historical transition, case study and interviewees.

The MLP distinguishes between three levels: micro-levels with (1) niche-innovations, meso-levels with (2) socio-technical regimes, and macro-levels with (3) socio-technical landscapes [38]. These levels of structuration can be defined according to Berkhout et al. [42] as:

(1)　Niches: ' . . . protected spaces for the development and use of promising technologies by means of experimentation, with the aim of learning about the desirability of the new technology, and enhancing the further development and the rate of application of the new technology' ([43], p. 186).

(2)　Regimes: ' . . . the rule set . . . embedded in a complex of engineering practices, production process technologies, product characteristics, skills and procedures, ways of handling relevant artefacts and persons, ways of defining problems; all of them embedded in institutions and infrastructures' ([44], p. 338).

(3)　Landscapes: 'the 'external environment' and consists of factors that not only affect the regime under analysis but a variety of other regimes as well [41] with ' . . . background variables such as the material infrastructure, political culture and coalitions, social values, worldviews and paradigms, the macro economy, demography and the natural environment which channel transition processes and change themselves slowly in an autonomous way' [41].

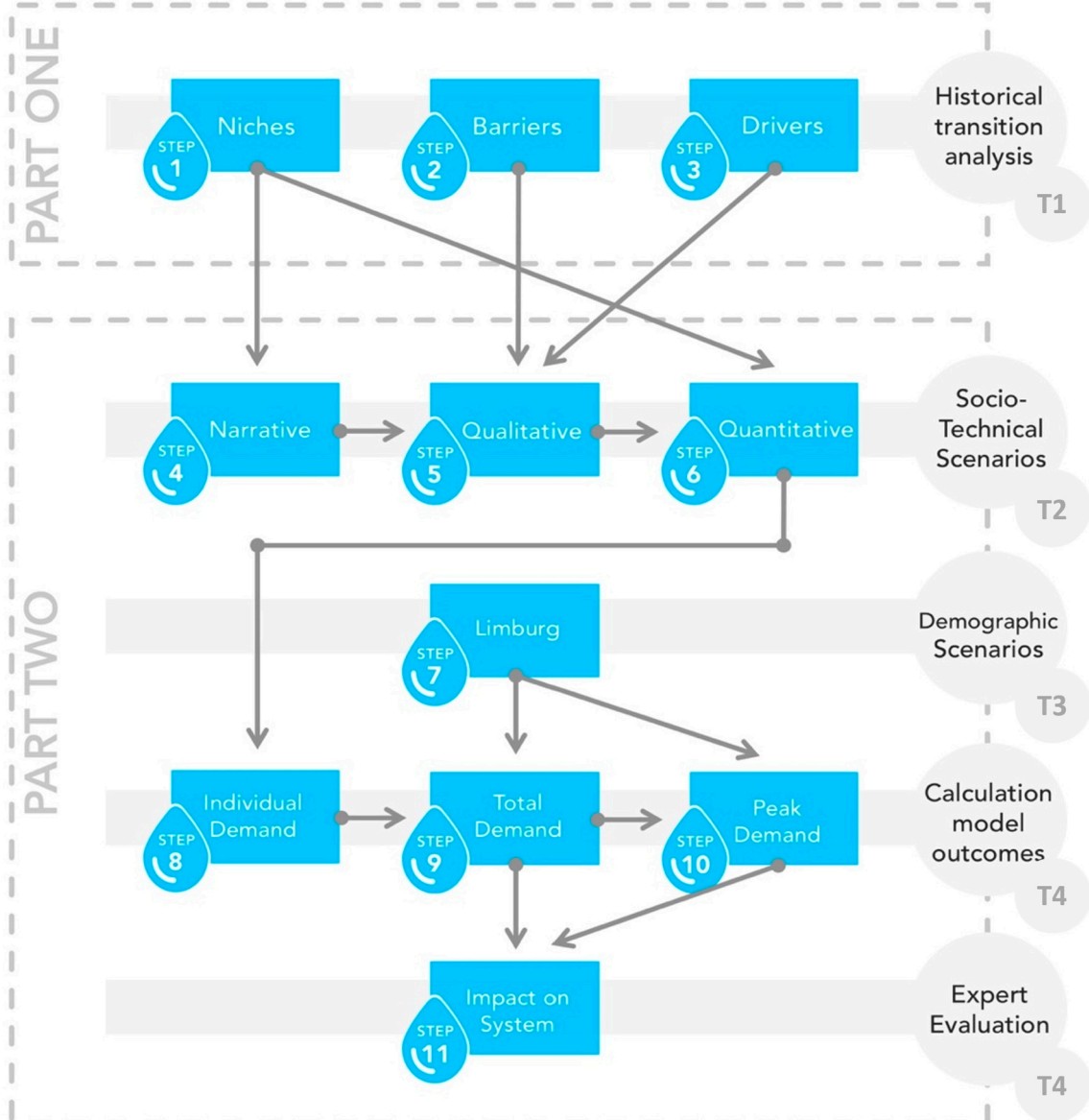

**Figure 2.** Conceptual framework.

Despite the wide application of the MLP in transition studies (also in the water sector) [1,6,45], it has also been critiqued. The research acknowledges the shortcomings of the framework (e.g., lack of addressing the role of agency [46] and power [47]). However, we find it applicable for use as a basis for scenario building in order to understand potential complex interactions. Moreover, a pragmatic approach has been applied through the use of mixed methods, in combination with qualitative and quantitative data sources (triangulation of data; Table 1).

**Table 1.** Research approach part one.

| Task/Step | Description | Further Explanation | Methods |
|---|---|---|---|
| **Task 1: Historical transition analysis** | Identify internal and external developments and aspects that (de)stimulate a transformation process from a centralised towards a more decentralised hybrid water supply system. | The term internal refers here to developments and aspects directly related to water sources, water provision and sewage systems. Indirect developments that influence the water system [41], such as population growth and climate change, are referred to as external developments. | Multi-level perspective. The SEPTED (social, economic, political, technological, environmental and demographic developments) approach is used for the external developments. |
| **STEP 1: Niches [43]** | Identify list of niches and detailed description | Different technologies and concepts, pilot projects, feasibility studies, research spaces and social platforms | Desk research, interviews with market parties and (pilot)projects |
| **STEP 2: Barriers** | Identify barriers for the upscaling of SUWM niches in the context of the current regime | A variety of barriers ranging from legislation, social frames, governance and technological issues | Literature review, semi-structured interviews both with actors inside and outside of the regime |
| **STEP 3: Drivers** | Identify drivers for upscaling of SUWM niches in the context of the current regime | A variety of drivers both external (landscape pressures) and internal (tensions within the regime) | Literature review, semi-structured interviews both with actors inside and outside of the regime |

Part one is based on literature, reports, news articles and information derived from 43 interviews in the Netherlands and Belgium (4 interviews). Some interviews were conducted with Belgian stakeholders due to the geographic position of this case study area (Figure 1), the similarities (e.g., large share of rural settlements and geographic parables), and a recent transformative process with regard to the large-scale introduction of rainwater systems. The interviewed stakeholders include market and technology developers (10), water utility employees (9), regional water authority employees (3), policymakers and lobbyists (3), residents that use decentralised water services (5), housing corporation developers (4), researchers (5) and project managers of decentralised projects (4). Finally, the scope of action of most interviewees were that of national (16) and provincial (14) level. Only the residents had a local scope of action, whilst some market and technology developers operated internationally.

The detailed description of the historical transition analysis is in Dutch and is not publicly disclosed [48]. A summary is provided in Section 3.1. This analysis forms the input for the scenarios developed and described in Section 3.2.

## 2.2. The Case Study: Foresight Study of Water Systems in Limburg

The foresight part of the study consists of three elements: (1) different socio-technical scenarios; (2) demographic scenarios in Limburg; and (3) an analysis which integrates the two previous elements. The methods used are described in Table 2.

To evaluate the potential of decentralised water technologies for the future, two qualitative scenarios were developed. A scenario approach was chosen due to the complex and uncertain nature at hand. Most forecasting methods focus on narrowly defined issues, and insufficiently consider the broader system [46] and its linkages. Scenarios allow for an integration of different developments and uncertainties. Moreover, it has been argued that by developing scenarios alongside stakeholders, there has been an increase in the acceptance of the content within these scenarios due to a sense of co-ownership [56]. Scenarios enable us to place the emergence of new technologies in the broader societal context of economic development, climate change, policy development, cultural and demographic developments. Furthermore, technologies should not be seen as independent from one another. Rather, it should be viewed as an interaction between competition, hybridisation, or complementary technologies. These complex interactions are often neglected in quantitative scenarios [41]; therefore, narratives and qualitative scenarios form the basis of the quantitative input [50] of this study.

A morphological scenario analysis has been applied which allows the combination of several key uncertainties [57] instead of the standard two axes [41,50]. The purpose of the scenarios was not to describe the most likely future state. However, since the scenarios serve a strategic goal for the water utility of Limburg (WML), the central question is to consider which is the most important scenario to focus on. In this case, the most important scenarios describe the future state with the highest potential for decentralised water technologies. Therefore, a set of parameters were chosen for each scenario and based on the most likely factors [57] which would enhance the upscale of decentralised technologies and systems, whilst still being considered plausible. The developed scenarios give insight under which circumstances—bundle of technologies, barriers and drivers—decentralised water systems (SUWM) increase in scale and market share.

In addition to the scenario storylines, three separate demographic scenarios were developed, describing different projections for population growth (or decline), and changes in the number of households and household composition. This was done because both population size and household size have an impact on potable water demand.

Additionally, an assessment was made based on the number of houses which will be newly built or substantially renovated, due to it being an opportunity to implement new water technologies. For instance, a vacuum toilet needs a pressured sewer system which most likely will not be installed within individual houses. Finally, the built area in Limburg is divided into two different categories, that of 'dense urban' and '(semi-) rural', representing different opportunities for water reuses systems, rainwater collection systems, and gardens which require space.

**Table 2.** Research approach part two.

| Number | Description | Further Explanation | Methods |
|---|---|---|---|
| **Task 2: Socio-technical scenarios** | Map the possible different scenarios for (centralised, decentralised or hybrid) water supply systems | What-If Scenarios that have a high impact but are still credible [40]. See Appendix A for overview of operationalisation (Table A1) | Morphological scenario analysis [49] based on STEP 1 to 3 |
| **STEP 4: Narrative** | Narrative story lines of future images | Two socio-technical scenarios. Circumstances are optimised for the SUWM niches | Coherent story of a world in which SUWM niches are integrated based on STEP 1 [50] |
| **STEP 5: Qualitative** | Qualitative input of drivers and barriers that underscore the narrative storylines | For each scenario, a mild and an extreme variant developed in which certain barriers are low while certain drivers are being emphasised | Based TRANSCE 3.0 method [41] for creating sustainability scenarios. Drivers (STEP 3) and barriers (STEP 2) 'entail influences existing in the environment of the system under study ([41], p. 102) |
| **STEP 6: Quantitative** | Quantitative input of different technologies | See Appendix A (Table A3) | Step 4 and 5 have been translated into quantitative inputs for different technologies |
| **Task 3: Demographic scenarios** | Define the different possible scenarios for the Province of Limburg for 2050 | Three demographic scenarios of Limburg on the level of neighbourhoods (n = 900). | Including data of population composition and change, housing developments, migration rates and so on. |
| **STEP 7: Limburg** | Three quantitative demographic scenarios for Limburg in 2050 | Scenario based on prognosis (Middle) and two outer boundary scenarios. | Based on regional studies carried out for the Province of Limburg [51–54]. For data input see Appendix A (Table A2) |
| **Task 4: Impact on centralised regime** | Determine the influence of decentralised SUWM on the current centralised regime in the Province of Limburg | Quantitative calculations of different scenario combinations (Table 3) analysed in relation to the current centralised infrastructure and regime | Data of STEP 6 and 7 have been merged in a Excel calculation model and analysed by an expert panel (distribution, hydrology, process technology, asset management, and strategy experts) |

| Number | Description | Further Explanation | Methods |
|---|---|---|---|
| **STEP 8: Individual demand** | Quantify the projected impact on the individual water demand | Average water demand per person per type of urbanity (Table A4) calculated for the two scenarios and its two variants | Based on quantitative input of STEP 6 and information of type of urbanity (STEP 7) |
| **STEP 9: Total demand** | Quantify the projected impact on the total water demand | Total water demand of the Province compared to the current total water demand and population changes | Individual water demands in STEP 8 have been combined with the demographic scenarios (STEP 7) |
| **STEP 10: Peak demand** | Quantify the projected impact on the peak water demand | Daily peak and hourly peak compared to the current situation in different regions (rural and urban) | Individual peak demand during peak moments (STEP 8) combined with demographic scenarios and climate studies [55] |
| **STEP 11: Impact on current system** | Specify the impact of the total water demand and peak demand on the functioning of the centralised water regime | Two extreme scenario—outer boundaries—combinations have been selected and further analysed by transdisciplinary expert panels | STEP 8 to 10 have been used as input for 3 expert sessions. They analysed and calculated what the potential impact would be of these scenarios for the current water provision system |

**Table 3.** Combined scenario outcomes (scenario combinations in italic are the selected pathways).

| Scenarios | Socio-Technical Scenario Let if Flow (LF) | | Socio-Technical Scenario Safe Water (SW) | |
|---|---|---|---|---|
| **Demographic Scenario Low** | Mild | Extreme | Mild | *Selected pathway Extreme* |
| **Demographic Scenario Middle** | Mild | Extreme | Mild | Extreme |
| **Demographic Scenario High** | Mild | *Selected pathway Extreme* | Mild | Extreme |

Demographic scenarios (STEP 7) have been merged with quantitative socio-technical scenarios (STEP 6) resulting in 12 different pathways (scenario combinations). The outcomes of the two most extreme pathways have been selected for further analysis by the group of transdisciplinary experts in STEP 11 (Table 3), in order to study the potential transformative changes and impact on the existing regime.

*2.3. Case Study Description*

The case study area is spread over 2209 km$^2$ and has 1.1 million inhabitants, of which 80% live in semi-urban and rural areas, and only 20% in dense urban areas. The water utility is a semi-public not-for-profit organisation which possesses a monopoly, granted by the national Government, for producing and delivering drinking water. The local sewage networks are managed by 31 different municipalities, transported and treated at wastewater treatments plants by the public wastewater utility which is part of the regional water authority. The wastewater is treated and discharged in rivers and creeks. Rainwater management in developed areas are the responsibility of municipalities, beyond urban settlements it becomes the task of the regional water authority, and finally, larger river systems are managed by the national water authority.

For water provision, the water utility relies on groundwater (70%) and surface water (30%) from the river the Meuse. Pressure has been placed on surface water due to low river levels and industrial contamination. There are, in total, 25 water production facilities which deliver around 70 million m$^3$ of water per year through a pipe network estimated to be around 9000 km. Additionally, 70% of water demand comes from private households, with the other 30% coming from industry and agriculture. On average, an individual uses 120 L per day [58]. The demand is slightly higher in urban areas; however, during peak hours the demand is seen to be higher in semi-urban and rural areas.

The northern region of the province is flat, whilst the south is characterised by hills and clay soils. Average annual rainfall depths of 750 mm are distributed evenly in the case study area, with a peak experienced in the summer season. The expected changes in climate include an increase in extreme rain events and longer droughts. Additionally, an increase of precipitation is expected during winter and a decrease in summer [59].

## 3. Analysis and Results

*3.1. Historical Transition Analysis*

3.1.1. Niches

The different types of niches identified in the Netherlands concerning decentralised water systems are (1) market niches, (2) social niches, (3) playgrounds such as pilot projects, and (4) research spaces for the development of promising sustainable technologies.

Based on the literature review, newspapers, and interviews, the following market niches have been selected as being the most promising when considering the context of the Netherlands. Firstly, rainwater systems are being promoted by governmental agencies and realised across the Netherlands. The main inconvenience of rainwater harvesting is the impossible nature of predicting reliable availabilities [2].

In turn, most of the interviewees did not view rainwater harvesting as a stand-alone solution for all water consumption. Therefore, the fit-for-purpose quality provision was considered most applicable for domestic non-potable applications. A connection with the central water system remains in most cases. The reason mentioned in interviews is that stormwater and wastewater peaks, due to severe weather events, are going to be solved on the household level. Secondly, dual-pipe systems (which includes grey water) on the household and neighbourhood scale have been a promising development in the Netherlands. However, due to an incidental failure—whereby several individuals succumbed to an illness due to consuming grey water. This was due to two pipes being exchanged, and, therefore, led to investments in dual-pipe systems being banned by the national government. This can be considered a backslash event [40], whereby a system failure in a niche project almost completely inhibits the niche development. In turn, strong regulations are still considered as a barrier for certain pilot projects [60]. With the rise of the circular economy, dual-pipe systems are regaining attention for resource recovery and, as a side effect, grey water reuse. Thirdly, water-saving devices are being developed and promoted. The household applications which require and use the largest share of water are, respectively, that of showers and toilets. Vacuum toilets (using only 1 litre per flush compared to 6 or 9) have received attention and have been introduced in several demonstration projects. With the aim to recover resources and energy, and a side benefit being the fact that less water is needed. A new generation of innovative showers are entering the market. In arid regions, fog showers have been developed with the aim of addressing water scarcity. Nonetheless, innovative showers which directly reuse the water (recirculation showers) are developed in the Netherlands with the aim of being more sustainable (in water and energy consumption), whilst increasing comfort. All have only recently entered the market (max. six years) and guarantee that comfort remains or even increases, whilst using 70–90% less water and energy. More comfort can be seen as an essential factor for customer demand when considering the increasing water demand of showering from 39.5 to 49.2 litres per person per day from 1992 till 2016, whilst the water use per minute for showers has not changed [58].

A diversity of social platforms (niches) have been promoted and introduced by governmental agencies and Non-Governmental Organisations have also been introduced, which range from serious games, information sites, student challenges, subsidy programmes, and larger movements such as the Amsterdam Rainproof programme: https://www.rainproof.nl/communication-material-in-english. These platforms raise awareness surrounding water problems and offer solutions for non-experts. What the actual impacts of these platforms are remains uncertain. Most important, is the increase in social platforms and subsidy programmes which stimulate decentralised water solutions.

Different pilot projects in the Netherlands exist which focus on decentralised water solutions. Some pilots have the aim of learning (research spaces), whilst others are projects with pioneers in living labs. What becomes clear is that sustainability is the main driver of these projects. Sustainability involves a diversity of aspects. Beyond water, elements include energy, recycling of materials, community development and wastewater (circularity and resource recovery). In fact, water is often not the primary focus of these projects [60]. Additionally, within these pilot projects, there is often a difference between plans and actual implementation. Different technologies are planned, but due to financial and legal barriers they are sometimes not implemented. Kieboom [61] argues that such demonstration projects often fail due to the complexity of such adaptive social systems being underestimated.

### 3.1.2. Barriers

The aforementioned technologies are generally not considered to be the limiting factor for upscaling [62]. The social–political environment in which the technologies need to be implemented within can be major barriers for change [62]. For example, the lack of institutionalisation (one of the drivers for upscaling). Therefore, the decentralised innovations have not been internalised as an option in the list of options for future users. Most of the housing corporations and citizens are not aware of the possibilities regarding decentralised water concepts. Seven key barriers for upscaling have been identified:

1. **Current centralised system incurs little failure:** The centralised system functions effectively and delivers high-quality water to everyone for a relatively low price and includes low health risks. Decentralised technologies are, therefore, not considered a solution for solving the failure issues of centralised potable water systems. The pressure on the existing infrastructure is limited and the future is uncertain. Actors outside the regime (e.g., market companies and researchers) emphasise the pitfalls and problems of the current system. Whilst, at the same time, regime players (e.g., water utilities and the national government) hold on to their existing system (e.g., monopoly) and point to current legislation, positive features, and label the current system as sustainable. This ambiguity stalls decentralised developments.

2. **Lack of individual business cases:** The niche technologies are expected to be more costly compared to those of the centralised system. Hence, a real business model for decentralised water is still missing when only focusing on water consumption [63–65]. Additionally, the centralised system requires a large infrastructural component, which can be seen as inflexible. Moreover, homes have been designed to suit a centralised system, through the use of a central input and output. This robust but inflexible socio-technical centralised system is a barrier for the upscale of SUWM measures. The costs for potable water mainly consist of fixed costs. As a consequence, a decrease of your potable water consumption does not lead to a proportional decrease in your water bills. Hence, a financial incentive to reduce water consumption is missing.

3. **Public health:** The current system has a low risk-level. Every improvement in areas such as sustainability, efficiency or price have little chance of acceptance if this would result in higher risks relating to public health, or even the perception that it would involve a higher risk [60]. Regarding decentralised systems, uncertainty exists in the Netherlands about the quality of the water sources and the quantity available to ensure a constant supply and safety of application [65].

4. **Legislation:** Rainwater and grey water use are limited by rules and regulations for large-scale projects (more than one household). Other aspects regarding rules and procedures are uncertain and are perceived as ambiguous by the interviewees. Examples of water-saving devices which have faced legal issues through the installation and use in public facilities, such as vacuum toilets and recirculation showers also exist. Legislation often leads to the continuation of the status quo. The rules and legislations are set in place for the functioning system, and these systems arrange themselves vice versa around the conditions of the set of rules and laws.

5. **Fragmented division of responsibility:** In collective decentralised systems, different types of responsibilities are prevalent. A new form of governance may be required to better facilitate a process of self-organisation. It was noted during several interviews that in the beginning of these projects, tasks were divided between a motivated group of residents and tasks carried out by external parties. However, as time passes, the ecologic ideology can weather away as residents move or pass away, and others join. Problems which may arise include, who is responsible in this new setting and friction between residents regarding consumption.

6. **Public acceptance:** A differentiation can be made between (1) actual public acceptance of different SUWM measures and household applications and (2) the opinion of investors and housing corporations about public acceptance of the different measures. The latter leads to lower acceptance levels according to the interviewees and is, therefore, a barrier for upscaling.

7. **Absence of collaboration:** The lack of collaboration between the regime and the niches makes the upscaling of decentralised technologies difficult within the current regime.

3.1.3. Drivers

Recently, regime players have been increasingly participating in decentralised pilot projects which can become a driver for upscaling, rather than a barrier. Seven key drivers have been identified:

1. **Population change:** Population decline could stimulate the introduction of decentralised systems. The centralised systems can be considered relatively costly for small groups of users in remote

places. Rainwater and wastewater can become an economically feasible alternative. Population growth in urban areas puts pressure on the existing water infrastructures and subsoil. Therefore, alternative water sources, local stormwater buffering and wastewater treatment are introduced in densely populated areas where the centralised infrastructure has reached its limits in the subsoil.

2. **Inclusive water prices and subsidies:** One of the reasons to reduce the water demand or find alternative sources is a price increase of tap water or the indirect use of it. Indirect uses are the wastewater discharge and the energy consumption for water uses which represent 25% of the energy used in a household. Subsidies are becoming more available in regard to stormwater runoff measures, which can be linked to water consumption when stored. Nonetheless, the barriers for homeowners are still relatively high due to the high costs (a business case lacks) [63–65]. However, the business-case calculations are based on one function—potable water costs—but do not internalise other aspects such as stormwater protection. A regime that includes a more holistic perspective in the total costs (energy, wastewater and externalities) could create room for a positive business case.

3. **Circular economy:** A fundamental shift that can result in water-saving devices and water reuse is the transition from a linear towards a circular economy. There is no blueprint for the circular economy, but the use of local sources, the local treatment of wastewater, water saving and the reuse of water is being researched and tested [60].

4. **Climate change adaptation:** Climate change adaptation forms an important driver for the introduction of decentralised SUWM measures, in particular, rainwater systems. The sewage systems in the Netherlands have been designed to discharge rain events of 20 mm up to 30 mm. The climate scenarios predict an increase in extreme rain events [59]. In relation, the stormwater management, regional water authorities and municipalities are responsible. The replacement of the sewage network has not been seen as a desirable solution. Municipalities can legally force (or stimulate through subsidies) households to manage the precipitation which falls on their own plot of land. Municipalities are increasingly using this form of regulation and subsidies to manage stormwater, forming a driver for the large-scale introduction of rainwater systems.

5. **Social values and autarky:** The need to be self-sufficient (autarky) and to act sustainably are noted as main drivers for decentralised systems in interviews, and in the study of van Alphen [60]. Which may be reinforced by the growing gap between society and politics, geopolitical conflicts and increasing attention for terrorist incidents. Additionally, the trust in institutions, organisations, and multinationals are declining [66], whilst environmental consciousness is a growing trend [67]. Furthermore, new urban planning forms create space for self-sufficient forms of living. In a study by Brouwer [68], observations were made that in the Netherlands, 65% of the water clients are interested in sustainability, of which 30% were willing to pay more if it were to be produced in a sustainable manner.

6. **Simplicity:** A prerequisite for upscaling is that the problem and solution are both understandable. For decentralised water systems, this is clearly the case according to the interviewees.

7. **Legislation:** Besides legal barriers, some changes which form a driver have also been noted. Firstly, the possibility of municipalities to distribute the responsibility of water management to households as described in point four. Secondly, a new environmental law ensuring that decentralised governments will have more freedom of policy to deliver local customised solutions. In principle, all types of decentralised developments are possible with the 'Yes, if' principle instead of the old 'No, unless' principle. Local decentralised solutions will find fewer legal barriers within this new law. Thirdly, in the past, a potable water connection was required for every household. This rule has recently been revised and withdrawn.

*3.2. Socio-Technical Scenarios*

Based on the barriers and drivers, two scenario storylines have been developed. The storylines do not represent the most likely scenarios. Instead, the focus is on developing plausible what-if-scenarios

with the largest potential for implementation of decentralised water technologies. This means that the scenarios will present the highest impact possible from decentralised technologies on the water utilities based on our current state of knowledge.

Consulting a group of experts, two sets of drivers where identified, leading to the diverging storylines. The first storyline, Let It Flow, is based on the following drivers:

- Strong focus on climate adaptation, reducing the effect of extreme precipitation
- Public campaigns and subsidies to collect and store rainwater
- Increase in green gardens
- Sustainability should not affect comfort and quality of life
- Less interest in energy saving due to abundance of renewables

The second storyline, Safe Water, is based on the following drivers:

- Strong focus on sustainability relating to water and energy saving
- Circular economy with a focus on reduction and reuse.
- Sustainable lifestyle is 'cool', showing off on social media
- Fast technological development of water-saving appliances.

The storylines are elaborated in Section 3.2.1. A more detailed list of the so-called SEPTED developments are given in the qualitative description (Section 3.2.2). For each scenario, two variants with different strengths were created: Mild and Extreme. The qualitative storylines were then translated into quantitative parameters, such as water use of different technologies and implementation rate of technologies (see Section 3.2.3). For an overview of the two types of scenarios, see Figure 3, which can be helpful for understanding the differences between the scenarios.

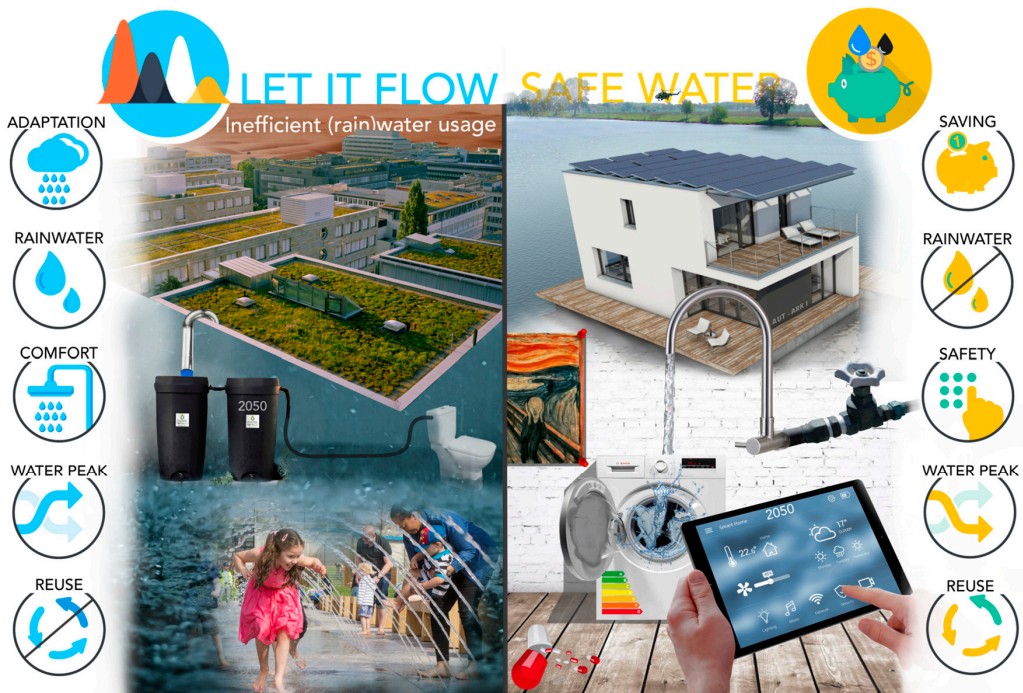

**Figure 3.** Illustration of the socio-technical scenarios: Let it Flow (**left**) and Safe Water (**right**).

3.2.1. Scenario Storylines

Let it Flow (LF)

In this scenario, society is willing to invest in sustainability as long as it does not lead to a decrease in comfort and quality of life. Through regulations, campaigns, and subsidies, citizens are encouraged

to capture and store rainwater and to use it in their households. It is free, soft for your laundry, and helps you maintain a pleasant and green garden. The main purpose of the measures is to reduce flood risk and keep stormwater out of the sewer system. Some people go as far as using the rainwater to shower, despite risks regarding water quality. Water saving at the household level does not take off. The abundance of rainwater seems to have the opposite effect: water just falls from the sky, so why not use it all?

For climate adaptation, an increase in private spaces converted into green gardens is needed to increase the infiltrating capacity, biodiversity and lower the urban heat island effect; these green gardens require extra watering during drier periods.

Unfortunately, during long periods of droughts, the storage capacities of households are empty. In these periods, when gardens are intensively watered, all water consumption needs to be delivered by tap water from the centralised supply system.

Safe Water (SW)

Society is willing to invest in large-scale saving of energy, costs, and water. Due to the changing climate such as droughts and more severe pollution events, aquifers and rivers become empty and are polluted at times. Scarcity in quality and quantity of water becomes a central issue. Limburg is not directly severely affected, but it does influence other developments. For instance, the government invests in campaigns, research funds and regulations (product standards) to save water and energy. One of the consequences is that wastewater is taxed per cubic metre instead of household composition. Hence, citizens are inclined to use less water or reuse water to reduce their wastewater, which results in a monetary saving. In the wider context, water scarcity is and remains the number one risk in the world. Therefore, Dutch companies invest in water-saving devices. Nevertheless, most citizens in the Netherlands are not necessarily interested in these technologies because of the water that is saved, but due to the energy bills being reduced. Still, the energy transition has left a large gap with heating water—which can be seen as its core leakage—which is filled with these efficient technologies. Circular technologies are often used to retain this previously lost energy. For instance, grey water is being reused to recover energy, but also to flush toilets as a win–win situation.

With the development of integrated solutions—e.g., using psychological and design principles—behaviour is optimised with the aim to save energy, water, and money. Many households have solar panels, which need to maintain an evenly distributed energy consumption pattern during the day. Smart technologies assist people in their domestic tasks (e.g., doing dishes, washing and watering the garden), even when residents are not at home.

The use of rainwater is a much more debated topic within this scenario. The risk perception of using rainwater for household purposes is rather negative. It holds substances such as pesticides, fine dust and other 'dirty' particles. Therefore, the use of rainwater is only fit for garden purposes. The Ministry of Infrastructure and Environment tries to keep the monopoly for the production of potable water with the water companies. We become more individualistic in this highly technocratic scenario and our social life happens to a large extent online. Green and maintained gardens have lost their appeal in this individualistic online world. The trend of paving front and backyards continues as it does nowadays. This means less watering during hot summer days.

3.2.2. Qualitative Scenarios

The storylines are explained in more detail in Table 4. The mild and extreme variants are described for different relevant aspects (left colon) for social, environmental, political, technological, and economic developments. The demographic developments are described in Section 3.3.

**Table 4.** Characteristics of the socio-technical scenarios: Let it flow (left) and Safe water (right).

| Characteristics | Socio-Technical Scenario Let it Flow (LF) | | Socio-Technical Scenario Safe Water (SW) | |
| --- | --- | --- | --- | --- |
| | **Mild** | **Extreme** | **Mild** | **Extreme** |
| **Social development** | *Quality of life.* Low tech in and around your house | *Quality of life.* Low tech in and around your house | *Technological individualism.* Smart and online | *Technological individualism.* Smart and online circularity |
| Sustainability discourse | 65% interested in sustainability, of which 30% accepts a price increase [68]. Use of rainwater is accepted but behavioural change not | Same as mild, but behavioural change only minor accepted | People hold a long-term view, so are willing to invest more. Behavioural changes are on an average level accepted | Same as mild but behavioural changes are widely accepted |
| Behaviour | People want comfort | People want even more comfort | People are willing to give a way some comfort for a new *minimalistic* lifestyle | People are willing to give a way comfort for a new *minimalistic* lifestyle |
| Social movements | Campaigns that highlight the need for using rainwater and climate adaptation | Campaigns that highlight the need for using rainwater and climate adaptation | Campaigns and movements that focus on circularity and lowering our footprint | Campaigns and movements that focus on circularity and lowering our footprint |
| Gardens | More green gardens for climate adaptive measures | Even more green gardens for climate adaptive measures | The trend of paving gardens keeps on growing. | The trend of paving gardens keeps on growing even more. |
| **Environmental development** | Scenario KNMI $W_L$ [59]: 898 mm/year precipitation. | Scenario KNMI $W_H$ [59]: 894 mm/year 250 mm in winter and 190 mm in summer precipitation. | Scenario KNMI $W_H$ [59]: 894 mm/year 250 mm in winter and 190 mm in summer precipitation. | Scenario KNMI $W_L$ [59]: 898 mm/year precipitation. |
| Weather | More extreme precipitation events but relatively well distributed over time. | More extreme precipitation events and longer periods of droughts. | More extreme precipitation events and longer periods of droughts. | More extreme precipitation events but relatively well distributed over time. |

**Table 4.** *Cont.*

| Characteristics | Socio-Technical Scenario Let it Flow (LF) | | Socio-Technical Scenario Safe Water (SW) | |
|---|---|---|---|---|
| | **Mild** | **Extreme** | **Mild** | **Extreme** |
| **Policy development** | Medium, aimed at private sector | High, aimed at private sector | Medium, aimed at market forces | High, aimed at market forces |
| Environmental policies | Medium, aimed at climate adaptation | High, aimed at climate adaptation | Medium, aimed at climate change prevention. Less energy consumption | High, aimed at climate change prevention. Less energy consumption |
| Legislation | New and renovated houses need to capture rainwater and us it for household purposes from 2030 | New and renovated houses need to capture rainwater and us it for household purposes from 2020. And infiltration in gardens | Rules and standards for products such as washing machines to lower their water and energy demand. New and renovated houses need to capture rainwater and us in for garden purposes | Rules and standards for products such as washing machines to lower their water and energy demand. New and renovated houses need to capture rainwater and us in for garden purposes |
| Subsidies | Regional water authorities give subsidies for rainwater decoupling | Same as mild, but also, more subsidies for existing houses to make them more climate adaptive. | Subsidies and tax deductions for new lease constructions for new energy and water saving household devices | Subsidies and tax deductions for new lease constructions for new energy and water saving household devices |
| Taxes | Low Value Added Taxes (VAT)-tariff on rainwater systems and possibility of tax deduction | Low VAT -tariff on rainwater systems and possibility of tax deduction | Wastewater tax per m$^3$ water instead of household size | Wastewater tax per m$^3$ water instead of household size |
| Risk acceptance | Risk acceptance for quality of water is medium but high for climate-related incidents (storms and flooding's) | Risk acceptance for quality of water is medium but high for climate-related incidents (storms and flooding's) | Acceptance of reuse of own water because people know what's inside. But low acceptance of rainwater use because it contains substances that are not controllable (fertilizers, herbicides and fine dust. | Acceptance of reuse of own water because people know what's inside. But low acceptance of rainwater use because it contains substances that are not controllable (fertilizers, herbicides and fine dust. |

**Table 4.** *Cont.*

| Characteristics | Socio-Technical Scenario Let it Flow (LF) | | Socio-Technical Scenario Safe Water (SW) | |
|---|---|---|---|---|
| | **Mild** | **Extreme** | **Mild** | **Extreme** |
| **Technological development** | Rainwater technologies mature. Improvements in water saving technologies stagnates | Same as mild. But also, improvements in renovation technologies in terms of possibilities and standardisation | Improvements in water saving technologies due to world water crisis | Improvements in water saving technologies due to world water crisis |
| Type of technological development | Low tech and biomimicry | Low tech and biomimicry | Efficiency and circularity | Efficiency and circularity |
| Price of technologies | Due to economies of scale the rainwater systems become cheaper in investment and installing costs | Same as mild. But also, the reinforcing feedback loop of the water price has a positive effect on the rainwater systems | Cost decrease of water and energy saving technologies and devices | Higher cost decrease of water and energy saving technologies and devices due to economies of scale |
| **Economic development** | Stabilisation (0% a year) and a levelling effect | Increase in economic growth (2% a year) but not levelled, more growth for the wealth [52]. Detached houses in rural areas have more economic possibilities | Stabilisation (0% a year) and a levelling effect | Increase in economic growth (2% a year) but not levelled, more growth for the wealthy [52]. Detached houses in rural areas have more economic possibilities |
| Price potable water | Same level | Over time, the water price increases due to rising costs of new infrastructure and a negative feedback loop | Same level, but energy prices increase | Over time the water price increases due to lower return for water companies although they have the same organisation and infrastructure. And energy prices increase |
| Investments | Investments in climate adaptive measures and comfort technologies | Investments in climate adaptive measures and comfort technologies | Investments in water saving technologies | Investments in water saving technologies |
| **Demographic development (STEP 7)** | Limburg High has the largest impact | Limburg High has the largest impact | Limburg Low has the largest impact | Limburg Low has the largest impact |

### 3.2.3. Quantitative Scenarios

The qualitative scenarios are translated into quantitative parameters such as behaviour, penetration rate of technologies, the usage capacity of technologies, sizes of gardens, precipitation patterns and number of persons per household. For the following technologies, quantitative inputs are selected: (1) toilet; (2) shower; (3) washing machine; (4) dishwasher; (5) gardens; (6) rainwater systems; (7) reuse of grey water. Different penetration rates have been selected for urban areas (🏢), rural areas (⛰), existing buildings, and renovated or constructed buildings (👷). The classification of types of buildings are described in Table A4 of Appendix A. The argumentation for certain percentages have been validated by the experts involved in the scenario building, based on literature [32–34,58,69–72], and data from market parties. The input data are listed in Table A3 of Appendix A.

### 3.3. Demographic Scenarios

The prognosis for Limburg 2050 [53] is taken as the middle scenario. Two outer demographic scenarios have been selected which deviate 10% from the middle scenario. In Table 5, general input data of the scenarios are presented. In the scenarios, regional differentiations are made up to neighbourhood level (900 neighbourhoods in Limburg).

**Table 5.** Demographic scenarios.

| Characteristics | Reference 2015 | Limburg Low 2050 | Limburg Middle 2050 | Limburg High 2050 |
|---|---|---|---|---|
| **Population** | 1,117,430 inhabitants<br>100% of 2015 | 869,376 inhabitants<br>77.80% of 2015 | 965,973 inhabitants<br>86.45% of 2015 | 1,062,570 inhabitants<br>95.09% of 2015 |
| **Average household size** | 2.2 | 2.1 | 2.1 | 2.1 |
| **Age** | | Higher average age | High average age | Medium average age |
| **Housing market** | 515,769 houses | 420,383<br>81.5% of 2015 | 467,093 houses<br>90.5% of 2015 | 513,802 houses<br>99.6% of 2015 |
| **Renovated/Newly build** | | 138,726 houses | 154,140 houses | 169,554 houses |
| **Corporation houses** | 144,725 houses | 117,959 houses | 131,066 houses | 144,173 houses |
| **Housing stock build before 1980** | 348,888 houses | - | - | - |

The current population dynamics are characterised by a decline and small shifts from rural areas towards cities. Firstly, increasing interest in urban areas (since the 90s), for cultural and economic reasons, and especially, the lack of urban areas in Limburg, results in a decline. Secondly, an ageing population without a decline in offspring results in a high death–birth ratio. At present, for every new child born, 1.36 persons dies [51]. Large regional differences in population changes are expected as well. The ageing population over time leads to a decline in water demand since the current 40+ generations use less water per capita compared to younger generations [58].

The number of households are still growing in Limburg, mainly due to the increase of single-household compositions, but it is expected to decline, with the tipping point occurring around 2022 [54]. However, not only the households in total but also the composition is of importance for water demand. A change is already occurring in the composition of households in Limburg towards a smaller number of people per household. The lowering number of people per household increases water demand because individuals in a single- or double-person household use, on average, more water than individuals in, for instance, a three- or four-person household [58].

Housing market developments are included in the demographic scenarios. A total of 16% of the population of Limburg lives in dense urban areas (see definition in Table A4 of Appendix A) and 84% in semi-urban and rural areas that include gardens. A differentiation is made between existing buildings and newly built or renovated buildings due to certain decentralised systems requiring more drastic measures. In the scenarios, 85% of the social housing stock will be new or renovated before

2050. Of the private sector, we assume that 20% of the housing stock older than 70 years in 2050 will be renovated or replaced.

## 3.4. Outcomes and Analysis

### 3.4.1. Total Water Demand

The input of the socio-technical scenarios results in the average water usage and demand, as shown in Figure 4. It is clear that for both the LF and the SW scenarios, the average water demand decreases, most being within newly built or renovated locations. The LF scenario has a significantly larger difference between the average consumption per person per day and tap water demand. This is the result of the large-scale introduction of rainwater systems in this scenario (64% of households). The difference in the SW scenario is lower because fewer households have a greywater reuse application (37%) compared to rainwater systems in the LF scenario, and this technology is often combined with water-saving devices so less water is needed to reuse (efficiency loss).

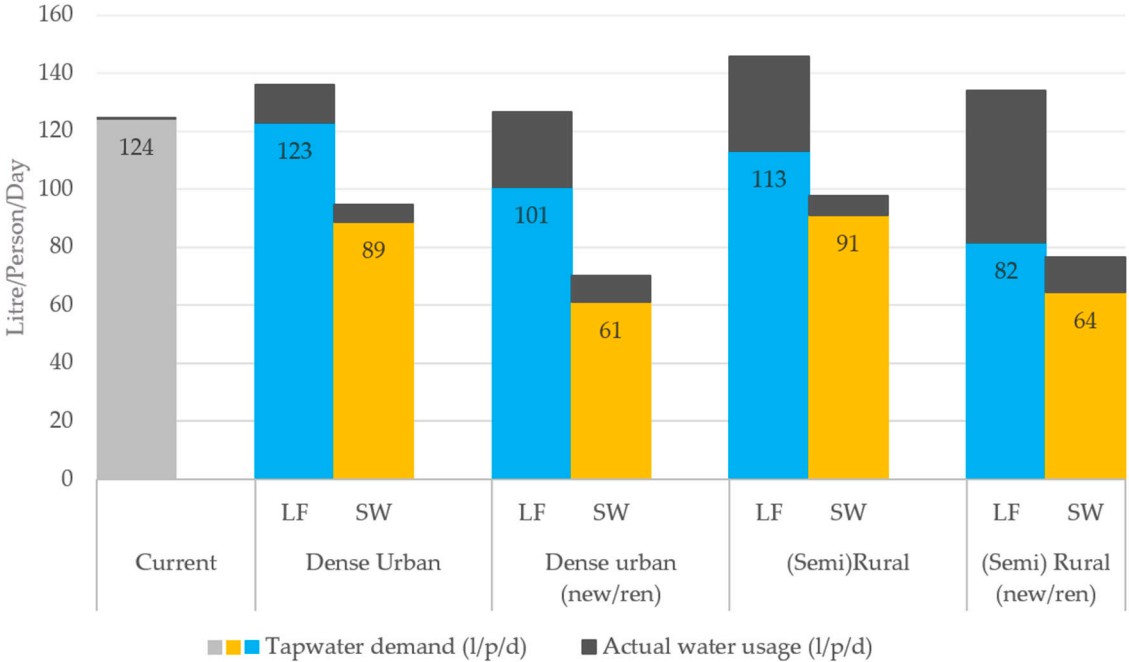

**Figure 4.** (Tap)water demand in litres per person per day (L/p/d) for both LF-extreme and SW-extreme.

The trend of the total household water demand for the entire province of Limburg is downward-oriented. Figure 5 shows that in all demographic scenarios, a decrease in the overall water demand can be expected. Firstly, due to a declining population (grey bars in Figure 5) with a maximum of 24% in the Limburg Low scenario. This is strengthened by the socio-technical scenarios. In the LF scenarios, the population change impacts are amplified by 12% and 15%. In the SW scenario, the water-saving and reusing technologies, behavioural changes and less-green gardens lead to a further decrease of around 26%. With a combined maximum of a 50% decrease in the Limburg Low and SW-extreme.

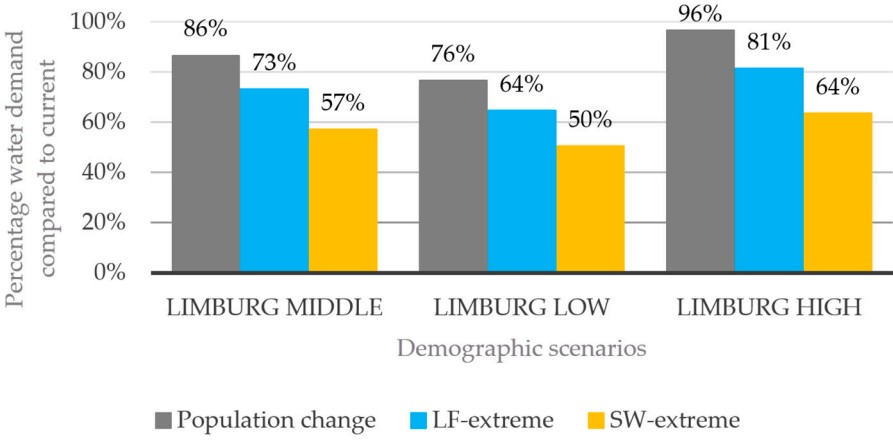

**Figure 5.** Change in total water demand in relation to the current demand (100%) in Limburg.

Differences in the average water demand between regions are the result of differences in population change and the rate of urbanity. In cluster South-East, the overall demand for all demographic scenarios is lower (between 8% and 10%) than in the cluster Middle. This is the result of the significantly larger population decline in the region South-East.

### 3.4.2. Peak Demand

The maximum day and maximum hour peaks in the water system appear during warm and dry summer periods. According to a study of Vonk et al. [55], the peaks will increase as a result of new climate situations. Especially, during dry periods having a correlated effect. In such periods, water is used for specific purposes. For instance, a large share of the population waters their garden, washes their car, fills their swimming pool, and takes an extra shower. The volumetric capacity of the infrastructure should be capable of dealing with the maximum hour peak. This peak happens in most areas between 7 and 10 PM. Especially, water use for outside (garden) applications, determining the yearly peak. In Limburg, a relatively large share of the population has a garden and, therefore, the peak is relatively high compared to other areas in the Netherlands. In dense urban areas, the consumption patterns are more evenly distributed. During the day, residents are often working in urban areas and, therefore, result in a smaller peak for urban areas. In rural areas, the early-morning shower peak (07:00) and watering garden evening peak (20:00) are visible (Figure 6).

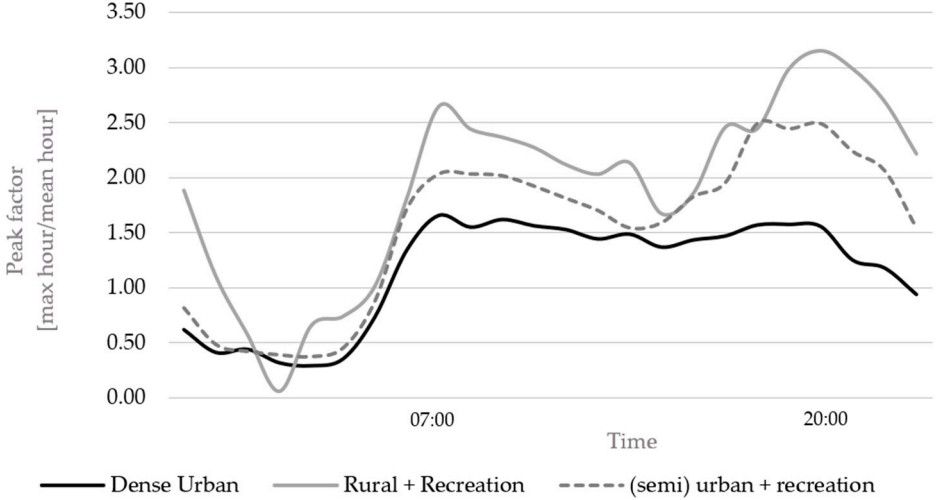

**Figure 6.** Historical peak factor distribution for different water provision clusters in Limburg (source: Waterleiding Maatschappij Limburg, The Netherlands).

In the LF scenario, there is an increase in the number of green gardens which need to be watered during dry spells. Furthermore, less or no water is available during dry periods in the rainwater tanks whilst the demand for this water increases. These tanks are empty because more household applications use this water on a daily basis (e.g., flush toilets). Lastly, people use more water in this scenario due to valuing comfort. Therefore, a larger increase in the peak occurs (see Figure 7: between 17:00 and 22:00). The difference between the average demand during a max-day (highest daily water demand of the year) and the hourly peak demand increases in the LF-extreme scenario compared to the current maximum daily demand, which can be seen by the steep curve of the LF-extreme scenario. This increase will not be evenly distributed in all areas. In dense urban areas, the peaks remain the same or decline, while in more rural and semi-urban areas, peaks increase even more than can be seen in Figure 7. Local differences, as presented in Figure 6, are amplified in the LF scenario while levelled in the SW scenario.

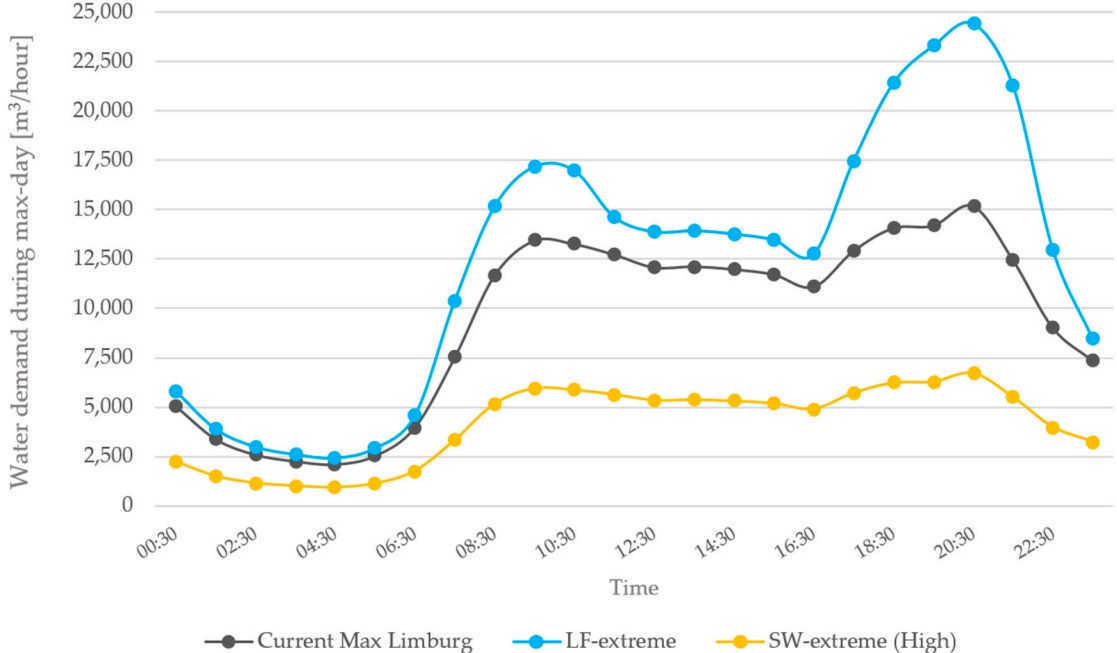

**Figure 7.** Water demand during max-day (peak) of Limburg for different socio-technical scenarios.

The daily maximum peak in the LF-extreme scenario is 34% higher than in the current situation (Figure 8: left). Regarding the hourly peak, the increase is even more severe, namely 61% compared to the current situation (Figure 8: right). Whereas, in the SW scenario, a decline is notable, even more than during the hourly peak (44% of current peak, Figure 8: right) compared to daily peak demands (50% of current demand, Figure 8: left). Fewer gardens need to be watered, this has a larger impact on the peak compared to the average demand due to irregular usage (during dry periods). Furthermore, rainwater is only used for gardens and not for in-house applications. Therefore, in drier periods, more water is available in tanks and rain barrels in the SW scenario than in the LF scenario.

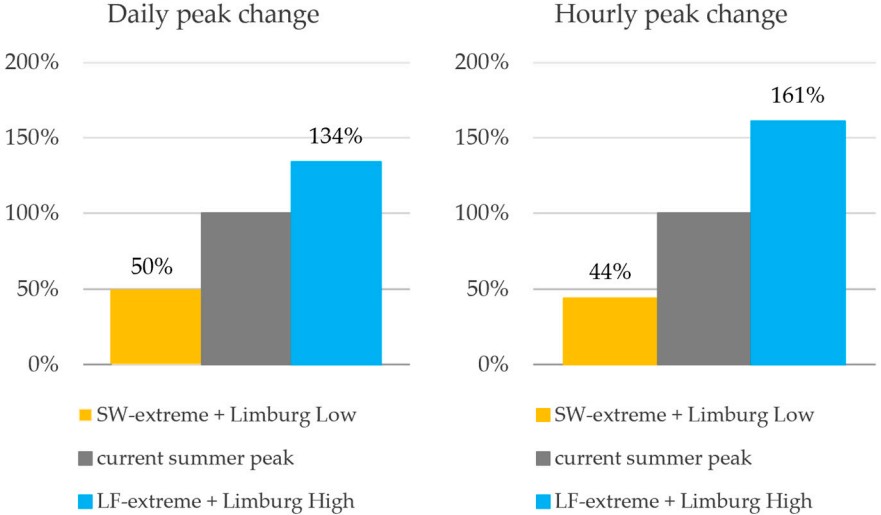

**Figure 8.** Change in summer day peak (**left**) and hour peak (**right**) in relation to current peak demands (100%) in Limburg.

### 3.5. Analysis of Impact of Scenarios on the Centralised System

The yearly water demand and peak water demand determine the functioning of the centralised potable water infrastructure. For instance, the total water demand represents half of the revenues of the water utility. The other half is based on fixed costs and are distributed over all connections (households), which is affected by the different demographic scenarios. Therefore, the water demand and the number of households influence the total revenue of the water utility. On the other hand, the daily peak demand determines the production and buffer capacity of the infrastructure. Furthermore, the hourly peak demand largely defines the design of the distribution network (pumps, pipes etc.).

#### 3.5.1. Impact of LF Scenario on Centralised System

The increase in the difference between the average water demand and the maximum peak demand creates an inefficient functioning organisation and configuration of the existing centralised asset system. The increase in the maximum hourly peak of the LF-extreme pathway results in extra investments in three domains of the water production system: (1) distribution; (2) purification; and (3) water catchments.

The first involves new water-storage capacities to ensure more buffering during the maximum day peaks. New high-pressure pumps are needed for the hourly peak. Additionally, an expansion and extension of the transport and the distribution network are needed. Secondly, for purification, more softening capacity is needed, which includes relatively high annual costs. The purification capacity of Limburg at the moment is large enough for the LF-extreme scenario and includes no extra investments. Thirdly, when groundwater is at its lowest level, the water extraction capacity of the water provider is one-third too little. Climate change does also affect the groundwater levels, which could further decrease the availability of fresh groundwater. Therefore, an assumption is made of installing an extra 70% water extraction capacity. New wells need to be drilled and installed. Therefore, new locations need to be financed and, at two locations, new purification plants including distribution pipelines need to be built (these are not yet taken into account in the financial calculation). The overall estimated investment costs are five times the yearly investment costs in infrastructure. The estimation has a fault margin of 50% and not all costs are included. For instance, a larger yearly capacity needs to be licensed, although it will not be used regularly.

Extra investment is not the only issue in the LF-extreme scenario. The extra available capacity will be rarely used, this results in several issues for the centralised system (overview in Table 6). Firstly, the rarely used capacity results in more maintenance. For example, due to an increase in the chance of

well clogging, which also reduces the level of assured service delivery. Secondly, climate change is likely to result in less groundwater replenishment. In Southern Limburg this can lead to problems during dry periods when the groundwater is low and the peaks (demand) are high. Shallow wells cannot be used during such periods. Hence, all new wells need to be rather deep, which includes extra investment costs. Thirdly, issues about the quality of water (mentioned by interviewees from Belgium) are the increased residence times of water [29] due to the volatile water demand. When a household rainwater storage capacity is empty, it switches back to the centralised grid. This water could have been stalled for a rather long period, with potential health risks. However, at the same moment, sediments that were deposited before, are potentially released and taken up in the water creating increased quality risks. This relates to the fifth issue: the lower demands during wet periods reduce the self-cleaning capacity of the centralised distribution network. Lastly, pump failures may occur more often due to the variable utilisation. All in all, the mentioned issues highlight that in the LF-extreme scenarios, the centralised system becomes less optimised than it currently is.

**Table 6.** Problems in the centralised system in the LF-extreme scenario.

| ISSUE | DESCRIPTION | EFFECTS |
|---|---|---|
| WELL CLOGGING | When wells are not frequently used (or in the same amount), there is an increased chance that they get constipated. | (1) More maintenance costs; (2) Decreased energy efficiency; (3) Decreased deliverability assurance. |
| BACTERIAL GROWTH | Bacteria can grow in filters and in the distribution network when there is an uneven burdening of for instance the filters due to the variable demands. | (1) More maintenance costs; (2) Increased health risks; (3) Increased quality issues. |
| SEDIMENT DEPOSITION | During periods of low demands (e.g., wet periods), sediment can be stored in pipe grid. When consumers switch back to the centralised system (e.g., dry period) the sediment can get mixed with the water. | (1) Increased health risks; (2) Lower quality of water (colour, smell etc.) (3) More maintenance costs, for flushing the pipes. |
| PUMP FAILURES | The chance of pump failures increases when they are less frequently used or on a lower intensity. | (1) Decreased deliverability assurance. (2) Extra pump capacity needed in reserve. |
| SELF-CLEANING CAPACITY | The self-cleaning capacity of the distribution system decreases with a lower average demand in the winter. This can result in sediment deposition and bacterial growth. | (1) More maintenance costs (2) More energy needed |
| WATER EXTRACTION | During peak moments (e.g., dry periods), the groundwater table is rather low—especially concerning climate change—while the water demand is high. | (1) Extra investment costs in deep wells. (2) Extra pump capacity needed in reserve. |
| RESIDENCE TIME | The same issue as sediment deposition only that due to the increased residence time of water the quality itself is affected and health issues may occur. | (1) Increased health risks; (2) Increased quality issues. |

The costs on the household level are not calculated for the scenario LF-extreme due to the lack of accurate data of extra costs. Nevertheless, the social costs, which are not calculated, are expected to increase. For an overview of these aspects see Table 7.

**Table 7.** Sustainability of centralised system in LF-extreme scenario.

| SUSTAINABLE | DESCRIPTION | PILLAR | |
| --- | --- | --- | --- |
| MORE MATERIALS NEEDED | More materials are needed but not used such as concrete, pumps etc. | Environmental and Economic | 👎 |
| MORE LAND NEEDED | More water extraction locations are needed, often in natural areas. | Environmental | 👎 |
| ENERGY and CLIMATE NEUTRAL | Difficult to become self-sufficient when the energy need is destabilised with a higher peak factor. Furthermore, energy efficiency reduces due to higher chance of failures and more energy is needed for pumping in deeper wells. | Environmental and Economic | 👎 |
| HOUSEHOLD COSTS | Expected increase in the costs for all households (fixed costs). | Social | 👎 |

To summarise, the analysis shows that the large-scale introduction of rainwater systems and green adaptation measures (green gardens) can have a severe impact on the existing potable water regime. It becomes less efficient, whilst at the same time, the number of failures will increase.

### 3.5.2. Impact of SW Scenario on Centralised System

The drastic decrease in the total water demand of around 50% results in higher water prices. Firstly, the costs related to the production facilities can only be reduced by 36% because the geographic distribution area remains the same size and only a few production locations can be closed. Secondly, the costs related to the distribution network stay more or less the same since most pipes will be needed in this scenario to deliver water in all regions of the province. Thirdly, the same counts for the total number of employees needed to keep the organisation functioning, this can only be reduced by a fraction compared to the water demand decrease of 50%. With a cost allocation model (internal model of water utility), the total costs of the organisation in 2050 were measured. The total costs for the water utility in the SW scenario are estimated to decrease by only 25%. Resulting in an increase of 51% in fluid costs (price charged per cubic meter water) for customers and an increase of 23% in fixed costs (price charged per connection) per household.

The impact on the yearly water bills for households is notable but may not be disruptive (Figure 9). A family of three that still has the same consumption pattern will see an increase in annual costs of €81 in 2050. A family that drastically reduces its consumption pattern (to around 50 L/p/d) will see a slight decrease in annual costs (€35). The difference between a three-person household that saves water and one that remains the same is around €116 in 2050. Hence, inequalities will increase. Wealthier households can invest more in water-saving devices which decrease their water bill but also decreases the efficiency of the water utility organisation. As a result, less wealthy individuals with no water-saving devices will see an increase in their annual bills. Lastly, this gap may even be increased

when looking at the energy bills which have crossovers with the water usage (warm water) and new stormwater and wastewater rules and subsidies.

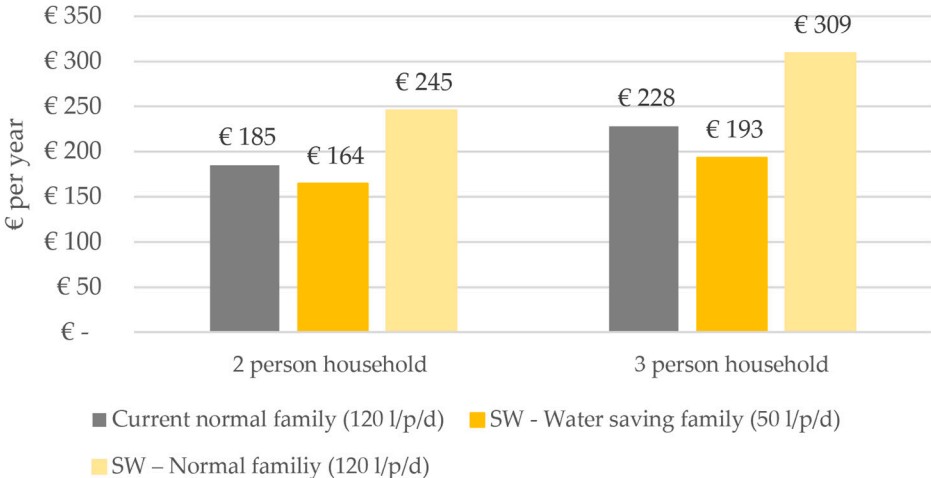

**Figure 9.** Annual prices for different households in the combined scenario SW-extreme and demographic low.

The SW scenario creates fewer problems for the functioning of the centralised infrastructure. A lower average demand peak ratio results in an optimised water provision system in which assets have a better utilisation rate. There is not an oversized system and this scenario, therefore, scores high on energy and material efficiency and fewer system failures. For an overview of sustainability change of the centralised system in the SW-extreme scenario see Table 8.

**Table 8.** Sustainability of centralised system in SW-extreme scenario.

| SUSTAINABLE | DESCRIPTION | PILLAR | |
| --- | --- | --- | --- |
| LESS MATERIALS NEEDED | Less materials are needed for the production due to the total water demand. And less assets are needed due to the lower peak. Materials for distribution stay the same. | Environmental and Economic | 👍 |
| LESS LAND NEEDED | Some water extraction locations can be closed and given back to nature. | Environmental | 👍 |
| ENERGY & CLIMATE NEUTRAL | Total energy demand decreases. But the energy used per m$^3$ water increases because the pressure of the total distribution system needs to stay the same with a lower deliverance. | Environmental and Economic | 👍👎 |
| HOUSEHOLD COSTS | Increase in the average costs for household and higher inequalities between water saving households and normal households (fluid costs). | Social | 👎 |

## 4. Discussion

Bias of the researchers which can be identified deals with the failure to take into account all of the possible variables. This bias makes qualitative research much more dependent on the judgement of experts [36]. Therefore, in the most critical parts (e.g., scenarios and outcomes), a large group of water experts were included. Expert opinions are added to long-term statistical data. On occasions when statistical data is lacking, such as for foresight, expert opinions are the substitution of the long-term data. Nonetheless, water and utility professionals with expertise of centralised services represented the majority; therefore, specific knowledge concerning decentralised systems might be overlooked. Water utility experts have been involved in the research design, analysis and evaluation, and the practical uptake.

Concerning the validity of the study, a lack of accurate data exists about sizes of gardens and the watering behaviour of households for gardens, as noted by others [55]. The input of these aspects has a large impact on the peak factor. For future research, information relating to watering behaviour is essential. For instance, including the role of climate change on watering behaviour. Although some data is rather uncertain or missing, most choices and assumptions were conservative (under-estimation of potential water use) to ensure the validity of this research. The impact of holidays and tourism have been neglected in this study, nonetheless, it can have both a damping and reinforcing impact on the (peak) demand [55]. We agree with Lucas et al. [73] that to design a water supply infrastructure, we should focus more on individual patterns rather than only on overall system patterns. This is especially the case when more decentralised water solutions are introduced. Local or individual peak demand are not analysed in this study, which can be considered a shortcoming. Nonetheless, the differences between different water provision clusters are significant but not a direct problem for the infrastructure in the case study. The drinking water utility is currently connecting all clusters of their distribution network. This optimisation increases the flexibility of the centralised grid because water shortages in one cluster can be filled with the extra capacity of other clusters. Increasing connectivity is mentioned as necessary by Leigh and Lee to become more resilient [11].

Speers and Mitchel [74], Arora et al. [30] and Lucas et al. [73] have found that decentralised SUWM measures can mitigate peak demands. However, this research has shown that the opposite can be the case when specific SUWM measures are implemented by one domain of the regime (e.g., stormwater). For example, decentralised rainwater systems are being introduced as sustainable solutions. The same counts for green gardens with the aim to reduce stormwater runoff, the urban heat island effect, and increase biodiversity. However, the effect that these measures have on the current centralised water provision system is rather unsustainable by increasing the peak demand whilst lowering the average demand as also found by Sapkota et al. [75]. It is recommended for future research to differentiate clearly between different types of SUWM measures and their related impacts. The literature focusses mainly on the positive aspects [11], but should also address potential negative feedback loops and externalities. These externalities can be overcome but should be addressed beforehand. Future research should focus more on interactions between different systems and scales by highlighting externalities and rebound effects. Even more so, future research should explore pathways to overcome such externalities and provide integrated multiple-problem and solution couplings for different SUWM measures. For one such measure, Lucas et al. [73] calculated that rainwater systems can decrease the peak demand, which is correct when water is available in rainwater tanks. Nonetheless, the LF scenario shows that this is not the case during dry spells. Market parties mentioned during interviews that they design the buffer capacity to supply rainwater 90% of the time. Hence, rainwater supplies are empty during dry spells, which is also included in urban water flow models [25]. Building larger storage capacities could be a technical solution but comes at higher economic and environmental costs [64]. However, economic calculation in future research should include costs related to expenditures of decentralised systems and their operations, as well as the avoided expenditures for water and stormwater infrastructures and operations.

The robust centralised water provision system in Limburg makes transformative changes or the ability to adapt difficult. Stranded assets and optimised systems have a damping effect on system changes. Comparable situations can be found across Europe where existing infrastructures are potentially inflexible for the effects of introduced SUWM measures. In other parts of the world where centralised systems are less developed or score lower on criteria low costs, safety, and assured-reliability, decentralised water systems may arise and grow even faster. Furthermore, Daniell et al. [24] mention that the uptake of decentralised water systems in more centralised multi-level governance systems, such as in China, can occur at a faster pace compared to more fragmented multi-governance systems with distributed power and resources, such as the Netherlands. Hence, more centralised multi-governance areas potentially face more drastic transformative processes. These more drastic transformations provide opportunities for sustainable solutions, as well as unforeseen rebound effects. The used methodology and topic can be applied for future research in different contexts and settings, providing valuable insights for science and practitioners.

This study has been limited in the scope by not including other consumers besides household. Industry, agriculture and recreation could have a damping or reinforcing impact on the scenarios and should be included in future studies. Nonetheless, household water consumption has a relative high share (70%) of the total public potable water demand in Limburg, which justifies the scope. Other shrinking regions which can be found in, for instance, Germany, Spain, and Czech Republic [76] have comparable household water shares (between 66% and 84%) of the public demand [77].

To validate the penetration rate of rainwater systems in the scenarios and the impact on the total water demand, a comparison is made with Flanders, Belgium. Since 2005, a Belgium law [78] ensures that for most of the new buildings and renovated projects, a rainwater storage capacity needs to be installed. The principle of this law is that at first, rainwater should be used as much as possible at the source (household). Thereafter, the remaining rainwater should be infiltrated or temporarily stored. In this way, the current sewage system is able to handle more intense rain events with the help of decentralised solutions. Based on a survey by the local utility, the penetration rate of rainwater systems is between 39% and 74% of the households in different provinces (excluding large cities). The introduction of rainwater systems in Belgium occurred at a more rapid pace compared to the developed scenarios of this study. The changed average potable water demand (from the centralised utility) per person in the Belgium province with a penetration rate of 39% has seen a decrease of 17% within 10 years. This change is more severe compared to the scenarios used in this article, which validates the probability of occurrence within this research. Nonetheless, information relating to comparable peak demands was missing, even though an increase was noticed in Belgium, as mentioned by the interviewees. The Flemish Government and water utilities prohibit the use of tap water for gardens during dry spells, lowering the peak demand. These measures are not yet taken in the Netherlands. Belgium can be used as an in-depth case study for future research to analyse the large-scale introduction of SUWM (rainwater systems) measures and the resulting impact on the existing centralised infrastructure. The framework developed by Sapkota et al. [75] can be used to analyse such hybrid water systems. Water balance models are used to simulate different scenarios in this framework such as the Urban Water Optioneering Tool (UWOT) [79] and the Urban Volume and Quality (UVQ) model [80]. If more context-specific information is available, geospatial models can be applied to derive more detailed conclusions. So far, this has only been done for virtual case studies [31] and it is, therefore, recommended to analyse real-world case studies.

The results have been presented to a diverse group of 20 practitioners (e.g., policymakers, strategist, market parties, lawyers, researchers etc.). They stated that policy-making in regard to climate adaptation and the circular economy is currently a debated topic, whilst drinking water is not addressed in the same debate. However, there are opportunities for joint solutions if a holistic approach is taken. Water utilities must, therefore, actively participate in the discussion and strive for coalition formation.

## 5. Conclusions

The historical transition analysis indicates that decentralised water niches are currently happening beneath the surface outside the potable water regime and waiting for a window of opportunity before they take-off in the Netherlands.

In the Let it Flow (LF) scenario, decentralised rainwater systems and green gardens will be used to solve a problem on one side of the regime (e.g., drainage for waste and stormwater), while it creates problems on the other side of the regime (e.g., lower utilisation rate, higher peaks, less efficiency, increased systems failures and health risks in the public potable water system). To conclude, the introduction of climate adaptation measures could have a disruptive effect on the centralised system in the LF scenario by making it less sustainable and optimal. It will not replace the centralised system because providing potable water will remain the task of the current organisations. Hence, a regime amidst diversification is the result.

In the Safe Water (SW) scenario, water-saving devices (e.g., recirculation showers and vacuum toilets) and dual-pipe systems with grey and black water streams are introduced to recover resources and reuse grey water. The impact on the centralised infrastructure is mainly positive (e.g., increased utilisation rate, lower peak factors and preservation of aquifers for future generations). The SW scenario has a sustaining effect on the centralised regime. However, the lower water demand makes the organisation, with stranded assets, as a whole, less efficient. Hence, prices for water will increase and result in more inequalities.

In conclusion, the large-scale implementation of different decentralised SUWM measures can have both positive and negative impacts on the current centralised system. Taking into account such impacts is key for transitioning SUWM into the dominant water regime. Currently, SUWM measures are still implemented in isolation and interactions in complex systems remain neglected.

**Author Contributions:** Conceptualization, D.v.D. and H.-J.v.A.; Data curation, D.v.D.; H-.J.v.A.; and S.H.A.K.; Formal analysis, D.v.D., Funding acquisition, H.-J.v.A. and E.d.B., Investigation: D.v.D.; H.-J.v.A.; S.H.A.K.; and E.d.B., Methodology: D.v.D. and H-.J.v.A., Project administration: H.-J.v.A. and E.d.B., Calculation: D.v.D., Supervision: E.d.B., Validation: D.v.D.; H.-J.v.A.; S.H.A.K.; and E.d.B., Visualisation: D.v.D., Writing—original draft: D.v.D., Writing—review & editing: S.H.A.K.; H.-J.v.A.; and E.d.B.

**Funding:** This paper was based on research financed by the joint research programme BTO that KWR Watercycle Research Institute carried out with researchers from the International Centre for Integrated assessment and Sustainable development (ICIS) in collaboration with water utilities WML and Dunea in the Netherlands.

**Conflicts of Interest:** The authors declare no conflicts of interest.

## Appendix A

**Table A1.** Socio-technical scenarios operationalisation.

| STEP | Socio-Technical Scenario | | Socio-Technical Scenario | |
|---|---|---|---|---|
| **Narrative (4)** | Let it flow | | Safe Water | |
| **Qualitative (5)** | Mild | Extreme | Mild | Extreme |
| **Quantitative (6)** | 1: Toilet<br>2: Shower<br>3: Dishwasher<br>4: Washing machine<br>5: Garden<br>6: Rainwater systems<br>7: Grey water systems | 1: Toilet<br>2: Shower<br>3: Dishwasher<br>4: Washing machine<br>5: Garden<br>6: Rainwater systems<br>7: Grey water systems | 1: Toilet<br>2: Shower<br>3: Dishwasher<br>4: Washing machine<br>5: Garden<br>6: Rainwater systems<br>7: Grey water systems | 1: Toilet<br>2: Shower<br>3: Dishwasher<br>4: Washing machine<br>5: Garden<br>6: Rainwater systems<br>7: Grey water systems |

**Table A2.** Input data for the demographic scenarios of Limburg (STEP 7).

| Characteristics | Demographic Scenario Low | Demographic Scenario Middle | Demographic Scenario High |
|---|---|---|---|
| **Population** | 1: Number of population<br>2. Number of newly born<br>3. Population composition | 1: Number of population<br>2. Number of newly born<br>3. Population composition | 1: Number of population<br>2. Number of newly born<br>3. Population composition |
| **Housing** | 1: Types of houses<br>2. Number of houses<br>3. Level of urbanity | 1: Types of houses<br>2. Number of houses<br>3. Level of urbanity | 1: Types of houses<br>2. Number of houses<br>3. Level of urbanity |
| **Developments** | 1: Number of new build<br>2. Number of renovations | 1: Number of new build<br>2. Number of renovations | 1: Number of new build<br>2. Number of renovations |
| **Regions** | 1. Region 1<br>1a. Sub region 1<br>1b. Sub-region 2<br>2. Region 2<br>3. Region 3 | 1. Region 1<br>1a. Sub region 1<br>1b. Sub-region 2<br>2. Region 2<br>3. Region 3 | 1. Region 1<br>1a. Sub region 1<br>1b. Sub-region 2<br>2. Region 2<br>3. Region 3 |

**Table A3.** Quantitative input data for individual water demands (STEP 6).

| TOILET | Current Situation | Let It Flow—Mild | | | | Let It Flow—Extreme | | | |
|---|---|---|---|---|---|---|---|---|---|
| Behaviour; times used per person per day (t/p/d) | 6 | 6 | 6 | 6 | 6 | 6 | 6 | 6 | 6 |
| Penetration flush blocker (%) | 73% | 80% | 100% | 80% | 100% | 80% | 100% | 80% | 100% |
| Percentage flush blocker used at a time (%) | 69% | 69% | 69% | 69% | 69% | 40% | 40% | 40% | 40% |
| Usage capacity flush blocker in litres at a time (L/t) | 3 | 3 | 3 | 3 | 3 | 3 | 3 | 3 | 3 |
| Penetration vacuum toilet and similar products (%) | 1% | 1% | 10% | 1% | 10% | 1% | 0% | 1% | 0% |
| Usage capacity vacuum toilet in litres at a time (L/t) | 1 | 1 | 1 | 1 | 1 | 1 | 1 | 1 | 1 |
| Average usage capacity regular toilet (L/t) | 7.90 | 7.70 | 6 | 7.70 | 6 | 7.20 | 6 | 7.70 | 6 |
| TOILET | Current Situation | Safe Water—Mild | | | | Safe Water—Extreme | | | |
| Behaviour; times used per person per day (t/p/d) | 6 | 6 | 6 | 6 | 6 | 6 | 6 | 6 | 6 |
| Penetration flush blocker (%) | 73% | 100% | 100% | 100% | 100% | 100% | 100% | 100% | 100% |
| Percentage flush blocker used at a time (%) | 69% | 69% | 69% | 69% | 69% | 69% | 69% | 69% | 69% |
| Usage capacity flush blocker in litres at a time (L/t) | 3 | 3 | 3 | 3 | 3 | 3 | 3 | 3 | 3 |
| Penetration vacuum toilet and similar products (%) | 1% | 1% | 25% | 1% | 20% | 1% | 50% | 1% | 40% |
| Usage capacity vacuum toilet in litres at a time (L/t) | 1 | 1 | 1 | 1 | 1 | 1 | 1 | 1 | 1 |
| Average usage capacity regular toilet (L/t) | 7.90 | 6 | 6 | 6 | 6 | 6 | 6 | 6 | 6 |

**Table A3.** *Cont.*

| | Current Situation | Let It Flow—Mild | | | | Let It Flow—Extreme | | | |
|---|---|---|---|---|---|---|---|---|---|
| SHOWER | | 🏢 | 👤🏢 | ⛰ | 👤⛰ | 🏢 | 👤🏢 | ⛰ | 👤⛰ |
| Average times showered a day per person (t/p/d) | 0.72 | 0.72 | 0.72 | 0.72 | 0.72 | 0.72 | 0.72 | 0.72 | 0.72 |
| Behaviour; minutes showered at a time (m/t) | 8.90 | 9.20 | 9.20 | 9.20 | 9.20 | 9.70 | 9.70 | 9.70 | 9.70 |
| Penetration water saving shower head (%) | 45% | 45% | 80% | 45% | 80% | 40% | 60% | 40% | 60% |
| Usage capacity water saving showerhead (L/m) | 7 | 7 | 7 | 7 | 7 | 7 | 7 | 7 | 7 |
| Penetration comfort shower (%) | 3% | 10% | 10% | 10% | 10% | 15% | 15% | 15% | 15% |
| Usage capacity comfort shower (L/m) | 15 | 15 | 15 | 15 | 15 | 15 | 15 | 15 | 15 |
| Penetration recycle and sprinkler shower (%) | 0% | 0% | 4% | 0% | 4% | 0% | 2% | 0% | 2% |
| Usage capacity recycle and sprinkler shower at a time (L/m) | 5 | 5 | 5 | 5 | 5 | 5 | 5 | 5 | 5 |
| Usage capacity regular shower per minute (L/m) | 8 | 8 | 8 | 8 | 8 | 8 | 8 | 8 | 8 |
| | Current Situation | Safe Water—Mild | | | | Safe Water—Extreme | | | |
| SHOWER | | 🏢 | 👤🏢 | ⛰ | 👤⛰ | 🏢 | 👤🏢 | ⛰ | 👤⛰ |
| Average times showered a day per person (t/p/d) | 0.72 | 0.72 | 0.72 | 0.72 | 0.72 | 0.72 | 0.72 | 0.72 | 0.72 |
| Behaviour; minutes showered at a time (m/t) | 8.90 | 8.60 | 8.60 | 8.60 | 8.60 | 8.10 | 8.10 | 8.10 | 8.10 |
| Penetration water saving shower head (%) | 45% | 50% | 83% | 50% | 83% | 60% | 70% | 60% | 70% |
| Usage capacity water saving showerhead (L/m) | 7 | 7 | 7 | 7 | 7 | 7 | 7 | 7 | 7 |
| Penetration comfort shower (%) | 3% | 5% | 2% | 5% | 2% | 4% | 0% | 4% | 0% |
| Usage capacity comfort shower (L/m) | 15 | 15 | 15 | 15 | 15 | 15 | 15 | 15 | 15 |
| Penetration recycle and sprinkler shower (%) | 0% | 3% | 15% | 3% | 15% | 7% | 30% | 7% | 30% |
| Usage capacity recycle and sprinkler shower at a time (L/m) | 5 | 5 | 5 | 5 | 5 | 5 | 5 | 5 | 5 |
| Usage capacity regular shower per minute (L/m) | 8 | 8 | 8 | 8 | 8 | 8 | 8 | 8 | 8 |

**Table A3.** *Cont.*

| | Current Situation | Let It Flow—Mild | | | | Let It Flow—Extreme | | | |
|---|---|---|---|---|---|---|---|---|---|
| WASHING | | | | | | | | | |
| Behaviour; times used per person per day (t/p/d) | 0.29 | 0.30 | 0.30 | 0.30 | 0.30 | 0.30 | 0.30 | 0.30 | 0.30 |
| Usage capacity regular washing machine in litres at a time (L/t) | 57 | 57 | 57 | 57 | 57 | 57 | 57 | 57 | 57 |
| Penetration semi-waterless washing machine (%) | 0% | 0% | 0% | 0% | 0% | 0% | 0% | 0% | 0% |
| Usage capacity semi-waterless washing machine in litres at a time (L/t) | 8.00 | 4.50 | 4.50 | 4.50 | 4.50 | 4.50 | 4.50 | 4.50 | 4.50 |
| | **Current Situation** | **Safe Water—Mild** | | | | **Safe Water—Extreme** | | | |
| WASHING | | | | | | | | | |
| Behaviour; times used per person per day (t/p/d) | 0.29 | 0.30 | 0.30 | 0.30 | 0.30 | 0.30 | 0.30 | 0.30 | 0.30 |
| Usage capacity regular washing machine in litres at a time (L/t) | 57 | 43 | 43 | 43 | 43 | 41 | 41 | 41 | 41 |
| Penetration semi-waterless washing machine (%) | 0% | 25% | 35% | 25% | 35% | 50% | 65% | 50% | 65% |
| Usage capacity semi-waterless washing machine in litres at a time (L/t) | 8.00 | 4.50 | 4.50 | 4.50 | 4.50 | 4.50 | 4.50 | 4.50 | 4.50 |
| | **Current Situation** | **Let It Flow—Mild** | | | | **Let It Flow—Extreme** | | | |
| DISHWASHER | | | | | | | | | |
| Penetration dishwasher (%) | 63% | 70% | 70% | 70% | 70% | 70% | 70% | 70% | 70% |
| Behaviour; times used per person per day (t/p/d) | 0.17 | 0.18 | 0.18 | 0.18 | 0.18 | 0.18 | 0.18 | 0.18 | 0.18 |
| Usage capacity regular dishwasher; litres at a times used (L/t) | 16 | 16 | 16 | 16 | 16 | 16 | 16 | 16 | 16 |
| Penetration for rinsing before dishwashing (%) | 38% | 38% | 38% | 38% | 38% | 38% | 38% | 38% | 38% |
| Usage capacity rinsing before dishwashing in litres at a time (L/t) | 7.50 | 7.50 | 7.50 | 7.50 | 7.50 | 7.50 | 7.50 | 7.50 | 7.50 |

**Table A3.** *Cont.*

| | Current Situation | Safe Water—Mild | | | | Safe Water—Extreme | | | |
|---|---|---|---|---|---|---|---|---|---|
| **DISHWASHER** | | | | | | | | | |
| Penetration dishwasher (%) | 63% | 70% | 70% | 70% | 70% | 70% | 70% | 70% | 70% |
| Behaviour; times used per person per day (t/p/d) | 0.17 | 0.16 | 0.16 | 0.16 | 0.16 | 0.16 | 0.16 | 0.16 | 0.16 |
| Usage capacity regular dishwasher; litres at a times used (L/t) | 16.00 | 10.50 | 10.50 | 10.50 | 10.50 | 9.50 | 9.50 | 9.50 | 9.50 |
| Penetration for rinsing before dishwashing (%) | 38% | 30% | 30% | 30% | 30% | 10% | 10% | 10% | 10% |
| Usage capacity rinsing before dishwashing in litres at a time (L/t) | 7.50 | 7.50 | 7.50 | 7.50 | 7.50 | 7.50 | 7.50 | 7.50 | 7.50 |

| | Current Situation | Let It Flow—Mild | | | | Let It Flow—Extreme | | | |
|---|---|---|---|---|---|---|---|---|---|
| **GARDENS** | | | | | | | | | |
| Penetration garden (%) | 76% | 30% | 30% | 100% | 100% | 30% | 30% | 100% | 100% |
| Penetration water capture and storage for garden (%) | 17% | 100% | 100% | 100% | 100% | 100% | 100% | 100% | 100% |
| Water availability in dry periods (%) | 40% | 40% | 30% | 40% | 30% | 30% | 5% | 30% | 5% |
| Sprinkled surface size in square meters (m$^2$) | 25 | 10 | 10 | 30 | 30 | 10 | 10 | 30 | 30 |
| Penetration paved surface / overgrown gardens (%) | 40% | 50% | 30% | 20% | 20% | 50% | 20% | 10% | 10% |
| Penetration vegetated gardens (%) | 60% | 50% | 70% | 80% | 80% | 50% | 80% | 90% | 90% |
| Number of times watering a year; during warm periods (t/y) | 20 | 21 | 21 | 21 | 21 | 20 | 20 | 20 | 20 |
| Number of times watering a year; during hot periods (t/y) | 3 | 3 | 3 | 3 | 3 | 4 | 4 | 4 | 4 |
| Millimetres watered during warm periods (mm/t) | 10 | 10 | 10 | 10 | 10 | 10 | 10 | 10 | 10 |
| Millimetres watered during hot periods (mm/t) | 15 | 15 | 15 | 15 | 15 | 15 | 15 | 15 | 15 |

Table A3. *Cont.*

| | Current Situation | Safe Water—Mild | | | | Safe Water—Extreme | | | |
|---|---|---|---|---|---|---|---|---|---|
| **GARDENS** | | 🏢 | 🏢 | 🏔 | 🏔 | 🏢 | 🏢 | 🏔 | 🏔 |
| Penetration garden (%) | 71% | 30% | 30% | 100% | 100% | 30% | 30% | 100% | 100% |
| Penetration water capture and storage for garden (%) | 17% | 70% | 70% | 80% | 80% | 70% | 70% | 80% | 80% |
| Water availability in dry periods (%) | 40% | 70% | 70% | 70% | 70% | 80% | 80% | 80% | 80% |
| Sprinkled surface size in square meters (m$^2$) | 25 | 10 | 10 | 30 | 30 | 10 | 10 | 30 | 30 |
| Penetration paved surface / overgrown gardens (%) | 40% | 60% | 50% | 40% | 40% | 70% | 70% | 60% | 60% |
| Penetration vegetated gardens (%) | 60% | 40% | 50% | 60% | 60% | 30% | 30% | 40% | 40% |
| Number of times watering a year; during warm periods (t/y) | 18 | 20 | 20 | 20 | 20 | 21 | 21 | 21 | 21 |
| Number of times watering a year; during hot periods (t/y) | 3 | 4 | 4 | 4 | 4 | 3 | 3 | 3 | 3 |
| Millimetres watered when above 20 °C (mm/t) | 10 | 10 | 10 | 10 | 10 | 10 | 10 | 10 | 10 |
| Millimetres watered when above 25 °C (mm/t) | 15 | 15 | 15 | 15 | 15 | 15 | 15 | 15 | 15 |
| | Current Situation | Let It Flow—Mild | | | | Let It Flow—Extreme | | | |
| **RAINWATER** | | 🏢 | 🏢 | 🏔 | 🏔 | 🏢 | 🏢 | 🏔 | 🏔 |
| Penetration rainwater capture and storage (%) | 12% | 50% | 60% | 70% | 80% | 100% | 100% | 100% | 100% |
| Penetration household applications linked to rainwater storage (%) | 0% | 5% | 50% | 15% | 50% | 30% | 100% | 50% | 100% |
| Useable rainwater in mm per day (mm/d) | 2 | 2 | 2 | 2 | 2 | 1.8 | 1.8 | 1.8 | 1.8 |
| Average roof size in square meter per household (m$^2$/h) | 57 | 50 | 50 | 65 | 65 | 50 | 50 | 65 | 65 |
| Average storage capacity in litres per household (L/h) | 500 | 2000 | 2000 | 5000 | 5000 | 3000 | 3000 | 7000 | 7000 |
| | Current Situation | Safe Water—Mild | | | | Safe Water—Extreme | | | |
| **REUSE** | | 🏢 | 🏢 | 🏔 | 🏔 | 🏢 | 🏢 | 🏔 | 🏔 |
| Penetration reuse system (%) | 0% | 10% | 30% | 10% | 30% | 20% | 70% | 20% | 70% |
| **LEGEND** | | | | | | | | | |
| L = litre t = times used m$^2$ = square metres mm = millimetres m = minute h = hour d = day y = year p = person % = percentage | | | | | | | | | |

**Table A4.** Types of buildings in scenarios.

| Icon | Classification | Description |
|:---:|:---:|:---:|
| 🏢 | Dense urban/Existing | Person living in existing (before 2020) building in area with more than 5000 inhabitants per square km |
| 🏢 | Dense urban/Renovated or new | Person living in renovated or new building in area with more than 5000 inhabitants per square km |
| ⛰ | (semi) Rural/Existing | Person living in existing (before 2020) building in area with less than 5000 inhabitants per square km |
| ⛰ | (semi) Rural/Renovated or new | Person living in renovated or new building in area with less than 5000 inhabitants per square km |

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
