# Peer review of "Potential Transformative Changes in Water Provision Systems: Impact of Decentralised Water Systems on Centralised Water Supply Regime"

_water, doi:10.3390/w11081709_

Round 1

Reviewer 1 Report

Dear Authors. The article presented deals with a very relevant topic, the methodology is very elaborate, but some parts are unclear and some information is missing. I hope to contribute to improve your work. I wish you much success.

Author Response

Notes are provided in the word document

Reviewer 2 Report

Manuscript deals with the crucial an very important issue concerning the impact of decentralised water systems on centralised water supply regime. The following answers should be referred to. Is this approach was consulted with water managers? What is the perspective of the future work? How in practice the results of the presented work can be used? This should be discussed in the point concerning discussion of the results. The OY axis should be described on the right side of the chart, as in Figure 7: Peak factor distribution of Limburg for different socio-technical scenarios, etc. According to the typographic rules, the decimal part is separated by a dot, and the comma is used, for example, to separate thousands, so the whole text, figures, tables should be prepared according to this rule, as for example in the Figure 6: Historical peak factor distribution for different water provision clusters in Limburg (source: WML).

Author Response

Notes are provided in the word document

Round 2

Reviewer 1 Report

Dear Authors. I have thought carefully about my past comments and the modifications made. I still consider that some aspects should be improved. But, I consider that the effort made is great and the theme of your work is of great relevance. I think it is necessary to publish works that can be used to improve the level of awareness and develop sustainable measures. Although I disagree with some minor aspects, in general I believe that your work can contribute to improving urban water management. Congratulations.

Author Response

Dear Reviewer,

Thank you very much for the constructive feedback, it had a positive influence on the quality of our article. 

You mentioned that still some aspects should be improved, but we haven't seen any new (restated) comments. So we interpret that the 'congratulations' means that your advice is to publish this work. We would like to thank you for this advice and we indeed hope that the outcomes of this study contribute to urban water management and the development of sustainable measures.  

Kind regards, 

Authors.

Reviewer 2 Report

Manuscript deals with the crucial an very important issue concerning the impact of decentralised water systems on centralised water supply regime. The following answers should be referred to. In section concerning discussion of results (in section of validation the penetration rate of rainwater systems in the scenarios and the impact on the total water demand …) add some information  about future perspectives of work, concerning the choice of reference, which should be supplemented with respect to the practical concept of water demand simulation, as in Pietrucha-Urbanik, K.; Studzinski, A. case study of failure simulation of pipelines conducted in chosen water supply system. Eksploat. Niezawodn. 2017, 19, 317–323. Section 5. Conslusions: What You mean by water saving devices and resource recovery technologies? In the Figure 7: Water demand during max-day (peak) of Limburg for different socio-technical scenarios The m3, 3 should be in the upper script.

Author Response

Dear Reviewer,

Thank you for the constructive feedback. The following questions have been adressed:

1. In the Figure 7: Water demand during max-day (peak) of Limburg for different socio-technical scenarios The m3, 3 should be in the upper script. 

The 3 has been changed to upper script in figure 7.

Conslusions: What You mean by water saving devices and resource recovery technologies? 

The sentence is changed to: 'In the Safe Water (SW) scenario, water saving devices (e.g. recirculation showers and vacuum toilets) and dual pipe systems with grey and black water streams are introduced to recover resources and reuse grey water.' For more detail see the socio-technical scenarios developed and table 11.

In section concerning discussion of results (in section of validation the penetration rate of rainwater systems in the scenarios and the impact on the total water demand …) add some information  about future perspectives of work, concerning the choice of reference, which should be supplemented with respect to the practical concept of water demand simulation, as in Pietrucha-Urbanik, K.; Studzinski, A. case study of failure simulation of pipelines conducted in chosen water supply system. Eksploat. Niezawodn. 2017, 19, 317–323. Section 5. 

For future research on water demand simulations the following suggestions have been made: 'The framework developed by Sapkota et al. [74] can be used to analyse such hybrid water systems. Water balance models are used to simulate different scenarios in the framework such as UWOT [78] and UVQ [79]. If more context specific information is available geospatial models can be applied to derive more detailed conclusions. So far this has only been done for virtual case studies [31] and it is therefore recommended to analyse real-world case studies.'

The study of Pietrucha-Urbanik and Studzinski is highly valuable but for this study too specific. We therefore did not include it in this article but we take it into account for future studies. For geospatial model case studies it can be a good method to zoom in into specific pipe failure issues. 

Thanks for the feedback and we hope we addressed all your questions.

Kind regards,

Authors.